

# Time-evolution of local information: Thermalization dynamics of local observables

**Thomas Klein Kvorning[1,2]⋆, Loïc Herviou[1,3] and Jens H. Bardarson[1]**

**1** Department of Physics, KTH Royal Institute of Technology, Stockholm, 106 91 Sweden
**2** Department of Physics, University of California, Berkeley, California 94720, USA
**3** Institute of Physics, Ecole Polytechnique Fédérale de Lausanne (EPFL),
CH-1015 Lausanne, Switzerland

⋆ kvorning@kth.se

## Abstract

Quantum many-body dynamics generically result in increasing entanglement that eventually leads to thermalization of local observables. This makes the exact description of the dynamics complex despite the apparent simplicity of (high-temperature) thermal states. For accurate but approximate simulations one needs a way to keep track of essential (quantum) information while discarding inessential one. To this end, we first introduce the concept of the information lattice, which supplements the physical spatial lattice with an additional dimension and where a local Hamiltonian gives rise to well-defined locally conserved von Neumann information current. This provides a convenient and insightful way of capturing the flow, through time and space, of information during quantum time-evolution, and gives a distinct signature of when local degrees of freedom decouple from long-range entanglement. As an example, we describe such decoupling of local degrees of freedom for the mixed-field transverse Ising model. Building on this, we secondly construct algorithms to time-evolve sets of local density matrices without any reference to a global state. With the notion of information currents, we motivate algorithms based on the intuition that information for statistical reasons flows from small to large scales. Using this guiding principle, we construct an algorithm that, at worst, shows two-digit convergence in time-evolutions up to very late times for diffusion process governed by the mixed-field transverse Ising Hamiltonian. While we focus on dynamics in 1D with nearest-neighbor Hamiltonians, the algorithms do not essentially rely on these assumptions and can in principle be generalized to higher dimensions and more complicated Hamiltonians.



# 1   Introduction

A numerical simulation of a many-body quantum system generally requires significantly more computational resources than its classical counterpart. This discrepancy is due to entanglement: a quantum state typically holds information that cannot be separated into sums of local parts, resulting in resources growing exponentially with the number of degrees of freedom. In equilibrium, the local nature of physical theories partially alleviates this problem, as thermal states generically have only short-range correlations [1–4]. Nevertheless, even for local theories, out-of-equilibrium time-evolution generically leads to a rapid buildup of correlations involving degrees of freedom spread over large scales [5].

Entanglement spread over large scales is, however, not directly observable. Instead, the set of density matrices of all small regions—*the local density matrices*—suffice to answer all physically relevant questions. In practice, measurement is mostly limited to either very few local degrees of freedom (e.g., single-spin polarization), or thermodynamic quantities such as the specific heat, susceptibilities to external fields, or transport properties such as heat and charge currents. Such quantities can generally be reframed as sums of local operators that act only within a small region and are therefore also captured by the local density matrices.

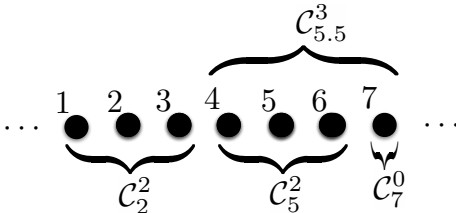

Figure 1: A one dimensional lattice with the lattice sites indexed by integers and a few line segments depicted. We let $\mathcal{C}_n^l$ denote the line segment with diameter $l$ centered at $n$. If a line segment contains an even number of sites, its center lies in-between two sites and $n$ is a half-integer, exemplified by $\mathcal{C}_{5.5}^3$ above.

To describe them requires resources increasing only linearly with system size, as opposed to exponentially for the entire wave function.

While entanglement buildup is unavoidable, there are reasons to believe that most of that entanglement does not influence the local density matrices. Generically, in the late-time steady state, the local density matrices coincide with those of a thermal density matrix, which one can predict without knowing any nonlocal degrees of freedom. This convergence can be understood in the following way: more quantum states have correlations on large scales than on small. Consequently, the information (a quantification of correlation) in small subsystems will, for statistical reasons, generically decrease, or equivalently, the entropy will increase—this accounts for the second law of thermodynamics for the entanglement entropy [6]. This statistical drift is the same for all scales, meaning that information will continue to flow to larger and larger scales, only bounded by system size. This motivates the guiding intuition of this article: generically, when information has reached a large enough scale, it will not flow back and affect local observables and can therefore be disregarded.

To utilize the idea that certain information can be disregarded, we need a way to specify where information is and to quantify how information flows. Only then can we know which information leaves the local (small) scales for good, and thus can be discarded. Unitary time-evolution implies that information is conserved, but it is fundamentally different from hydrodynamic conserved quantities such as energy. If we have a local Hamiltonian, energy is a "substance" in the sense that we have a well-defined notion of where it is and how it flows. The same cannot be said for information: because of the existence of non-local degrees of freedom, there is no well-defined notion of *where* information is located. To remedy this problem we introduce in this article a way to organize information into a local structure which we dub the *information lattice*. In the information lattice the physical space is supplemented with an extra dimension, quantifying how spread out the information is, thereby allowing information to be treated as a locally conserved quantity. The decomposition of information on the information lattice is the primary tool we will consequently use to analyze quantum dynamics, and is discussed in detail in section II.

In the context of quench dynamics, we ask in section III the question: can one tell from the information distribution how and when the local density matrices decouple from long-range correlations? When a system reaches equilibrium, or more exotically, when it approaches a state with localized excitations bouncing around as billiard balls, then we can show from the information distribution that there is an exact (and numerically easy to implement) decoupling of the local observables. In these situations we find an information gap, a range of scales with no information, which implies a decoupling of the local density matrices from long-range correlations.

Unfortunately, an information gap does not appear (in a finite time) in a generic setting. Nevertheless, one can simply try to time-evolve the local density matrices using some trun-

cation [7–13]. The general idea is as follows: for each time-step $\delta t$, the evolution of the density state $\rho$ of the entire system can be decomposed into two pieces. First the exact $\delta t$ time-evolution is performed and then $\rho$ is truncated by some function $T$, $\rho \to T(\rho)$, designed such that a part of the local observables is preserved (exactly what is preserved varies [7–13]). As the truncation variable increases more and more local degrees of freedom are preserved, and the exact time-evolution is recovered when the truncation variable is taken to infinity. If the time-evolution of the local observables converges at a finite value as one increases the truncation, it is plausible that one has captured the true time-evolution. This article's guiding principle—when information has reached a large enough scale, it will not come back— motivates why such a truncation scheme could work: if the error is introduced on a large enough scale, these erroneous correlations will propagate to larger scales and not affect the local observables.

A detailed analysis of specific quench dynamics reveals what can go wrong in such an approach. In the most straightforward truncation scheme with the mentioned properties, $T$ transforms the state $\rho$ into a state with correlations decaying exponentially with scale, and the decay length $\lambda$ increases with the truncation variable. Such an approximation, unfortunately, generically leads to a systematic underestimation of the information *flow* at scales $\sim \lambda$, leading to a buildup of erroneous correlations at scales $\sim \lambda$. Even if information generically flows from smaller to larger scales, if the erroneous correlations become significant, only a tiny fraction of it returning to small scales would alter the time-evolution of the local density matrices. To remedy the underestimation of the information current we require an additional property of $T$: it should both preserve the local observables and accurately approximate the information current out of the smallest scales. The second main result of our work is to construct an algorithm based on this idea. This algorithm shows good convergence properties and thus provides an example of how analyzing dynamics using the information lattice can lead to valuable insights on simulating quantum dynamics efficiently.

To summarize, in this paper, we construct the information lattice, a way to quantify where information is and how it flows. We present it with the required information-theoretic background in section II. From the information distribution, one can directly derive a decoupling of the local observables under certain circumstances. When one cannot, by analyzing quantum quench dynamics using the information lattice, it becomes evident that the most direct algorithms trying to utilize a decoupling of the local observables will, at some scale, underestimate the information current and can therefore readily be improved. We present this analysis of quantum quench dynamics using the information lattice in section III. Finally, in section IV, we construct an algorithm that implements the correct information flow, and in section V, we analyze its convergence properties.

## 2 The information lattice

To discuss quantum dynamics in terms of where information is located and how it flows, we need to quantify these notions, and to this end we introduce *the information lattice*. To define it, we first need to review the concept of total information in a quantum state. Intuitively, the total information in a quantum state should quantify how much one can predict knowing the whole state via the density matrix $\rho$. The von Neumann information, the deficit of the von Neumann entropy $S(\rho)$ [14] from its maximum,

$$I(\rho) = \log_2[\dim(\rho)] - S(\rho) = \log_2[\dim(\rho)] + \mathrm{Tr}[\rho \log_2(\rho)], \tag{1}$$

gives a precise meaning to this intuition. To understand it, consider a state $\rho$ which is a product state of maximally mixed states on all sites except one, where it gives a statistical prediction on

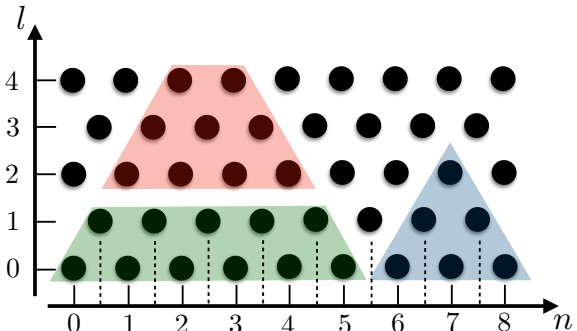

Figure 2: Each point $(n, l)$ in the information lattice correspond to a line segment $\mathcal{C}_n^l = [n - l/2, n + l/2]$. The value at a point is the information in the corresponding line segment that cannot be found on any smaller scale. Every triangle for which $(n, l)$ is the top consists of points that correspond to line segment subsets of $\mathcal{C}_n^l$. Therefore, summing all values in a triangle with base at $l = 0$ adds up to the total information in the line segment corresponding to the top of the triangle. As an example, summing the values in the blue triangle gives the total information in $\mathcal{C}_7^2 = [6, 8]$. If $i_n^l$ is zero in some region, such as the red region in the top left, the density matrices in that region can be reconstructed from smaller density matrices corresponding to the green region in the bottom left.

a single yes/no measurement. If $\rho$ predicted with certainty the outcome of this measurement, we could with $\rho$ answer exactly one yes/no question. Thus, $\rho$ would provide a single bit of information. With the conventions implied by the definition of entropy (1) a bit of information is given the value 1 [1]. If $\rho$ instead only gives a probability for the different outcomes, then $\rho$ does not provide a definite prediction to any observation. Repeating the measurement a significant number of times, one gets a well-defined average number of bits per measurement, $k$, needed to reproduce the string of outcomes [15]. The state can, thus, on average provide at most (measuring in a suitable base) $1 - k$ bits per measurement. There is thus in this average sense $1 - k$ bits of information in the system. So the von Neumann information is in this case $(1 - k)$.

In general the von Neuman information is the total information in a state in the average sense from the previous example. Depending on the measurement one can, knowing the full density matrix, predict different amount of information about the measurement outcomes. The von Neumann information is the maximum average number of bits one could predict. Similarly, the von Neumann information,

$$I_\rho(A) = \log_2[\dim(\rho_A)] - S(\rho_A), \tag{2}$$

of a reduced density matrix,

$$\rho_A = \mathop{\mathrm{Tr}}_{A^c} \rho, \quad A^c = A \text{ complement}, \tag{3}$$

on a region $A$ provides the information in $A$, quantifying how many observables in $A$ can be predicted from knowing $\rho$.

We define the 1D information lattice as the decomposition of the total information, $I(\rho) = \sum_{\mathcal{C}} i_{\mathcal{C}}(\rho)$, into the irreducible information $i_{\mathcal{C}}(\rho)$ on all possible continuous line segments $\mathcal{C}$. Specifically, $i_{\mathcal{C}}(\rho)$ is the information in $\rho_{\mathcal{C}}$ not contained in any $\rho_{\tilde{\mathcal{C}}}$ on a line segment $\tilde{\mathcal{C}}$ that is a proper subset of $\mathcal{C}$ (from now on we often refer to the information in a line segment

---

[1] One can change the convention of giving a single bit the value 1 by choosing another logarithm base in the definition of entropy (1).

$\mathcal{C}$ as a shorthand for the information in the reduced density matrix $\rho_{\mathcal{C}}$ of the line segment.) $i_{\mathcal{C}}(\rho)$ quantifies what the reduced density matrix $\rho_{\mathcal{C}}$ can predict which cannot also be predicted by the set of reduced density matrices of the proper subset line segments: $\{\rho_{\tilde{\mathcal{C}}}\}_{\tilde{\mathcal{C}} \subset \mathcal{C}}$. The information lattice can be generalized to arbitrary dimensions by letting $\mathcal{C}$ run over connected clusters instead of line segments. However, the expressions for $i_{\mathcal{C}}(\rho)$ in higher dimensions do not take forms as simple as they do in $1D$; we leave such higher-dimensional generalisations to future work.

The set of line segments is naturally organized into a $2D$ lattice (motivating the name information *lattice*), and on *this* lattice the decomposition makes information reminiscent of a hydrodynamic conserved quantity with well-defined local densities and currents. We label the line segments by their location $n$ and diameter $l$ (which we also refer to as scale),

$$\mathcal{C}_n^l = [n - l/2, n + l/2], \tag{4}$$

$n$ is an integer if $l$ is even, half-integer if $l$ is odd, see Fig. 1. The lattice sites are labeled by these indices and naturally take the form of a $2D$ lattice. Every triangle with $(n, l)$ at the top and base at scale $l = 0$ consists of points that correspond to line segment subsets of $\mathcal{C}_n^l$, see Fig. 2. Therefore, summing all values in a triangle with base at $l = 0$ adds up to the total information in the density matrix corresponding to the top of the triangle

To translate these definitions of $i_n^l$ into explicit expression we begin with $l = 0$. Since $\mathcal{C}_n^0$ has no proper subset line segment, the information in $\mathcal{C}_n^0$ not also present in any subset, is simply the total information on site $n$,

$$i_n^0 = I(\mathcal{C}_n^0). \tag{5}$$

For the definition for $l = 1$ we require the concept of the mutual information $I_\rho(A; B)$ between two disjoint regions $A$ and $B$. This is defined as the information in $AB = A \cup B$ that is neither in $A$ nor in $B$,

$$I_\rho(A; B) = I(\rho_{AB}) - I(\rho_A) - I(\rho_B). \tag{6}$$

With this, $i_n^{l=1}$, the information in $[n - 1/2, n + 1/2]$ not also present on site $n - 1/2$ or $n + 1/2$, is just the mutual information between the two sites,

$$i_n^1 = I(\mathcal{C}_{n-1/2}^0; \mathcal{C}_{n+1/2}^0). \tag{7}$$

To define $i_n^l$ for $l > 1$, we generalize $I_\rho(A; B)$[2] to *overlapping* sets. To get the expression for $I_\rho(A; B)$ we take the full information in $AB$, subtract the information in $A$ and $B$ separately, and add back the information in $A \cap B$, since otherwise it is subtracted twice,

$$I_\rho(A; B) = I(\rho_{AB}) - I(\rho_A) - I(\rho_B) + I(\rho_{A \cap B}). \tag{8}$$

One might worry that the information is not additive in the way assumed by these subtractions: what if there are situations when one could predict some observables not in $A \cap B$ but in $B$, from the density matrix on $\rho_A$? Then, apart from the information in $A \cap B$ (already corrected for), there could be information counted both in $I(\rho_A)$ and $I(\rho_B)$, and thus subtracted twice in (8). However, that would mean that there would exist some state $\rho$ where the expression (8) is negative, which is not the case [16, 17]. Information thus has the assumed additive property (strong subadditivity) and the expression (8) is correct. With $I_\rho(A; B)$ defined also for overlapping sets we have expressions for all the information lattice values,

$$i_n^l = \begin{cases} I(\mathcal{C}_n^l), & l = 0, \\ I(\mathcal{C}_{n-1/2}^{l-1}; \mathcal{C}_{n+1/2}^{l-1}), & l > 0. \end{cases} \tag{9}$$

---

[2]The expression in (8) is more conventionally denoted $I_\rho(A \setminus B; B \setminus A | A \cup B)$, since it can be interpreted as the mutual information between $A \setminus B$ (the part of $A$ not in $B$) and $B \setminus A$, conditioned on the intersection $A \cap B$.

## 2.1 Reconstruction of density matrices from subsets

One interpretation of a reduced density matrix $\rho_{AB}$ is as an encoding of the knowledge of the value of all observables on $AB$. If most information about the values of observables on $AB$ is known already from the subsets $A$ and $B$, then the density matrix $\rho_{AB}$ can be approximated from the density matrices of the subsets. For small $I_\rho(A;B)$ there are several ways to approximately construct $\rho_{AB}$ from $\rho_A$ and $\rho_B$ [18]. Of these, the twisted Petz recovery map [18]

$$\Phi^{\text{TPRM}}(\rho_A, \rho_B) = \exp\left(\ln \rho_A + \ln \rho_B - \ln \rho_{A \cap B}\right), \tag{10}$$

has a known bound [19] on the error,

$$\text{Tr} \sqrt{(\rho_{AB} - \Phi^{\text{TPRM}}(\rho_A, \rho_B))^2} \le 2\sqrt{I_\rho(A;B)}, \tag{11}$$

stating that for given $I_\rho(A;B)$ and reduced density matrices $\rho_A$ and $\rho_B$, all density matrices $\rho_{AB}$ must be within a radius $2\sqrt{I_\rho(A;B)}$ trace-norm ball centered at $\Phi^{\text{TPRM}}(\rho_A, \rho_B)$. Turning to the information lattice, this means that if the information in a region (e.g., the red region in Fig. 2) is small, then one can reconstruct the corresponding density matrices from the density matrices corresponding to the information lattice values below (such as the green region in Fig. 2).

## 2.2 Summation of information lattice values

From our definition of $i_n^l$ it follows that the information lattice values in a triangle with base at $l = 0$ sum up to the total information corresponding to the line segment at the tip of the triangle,

$$I(\mathcal{C}_n^l) = \sum_{(l', n') \in S_n^l} i_{n'}^{l'}, \qquad S_n^l = \{(l', n') | \mathcal{C}_{n'}^{l'} \subseteq \mathcal{C}_n^l\}. \tag{12}$$

To be consistent, this should also follow from the analytical definition in Eq. (9). We show this in general below, but to gain intuition we first consider a few specific examples to see how the sum over the entire lattice,

$$I(\rho) = \sum_{\text{all } n,l} i_n^l, \tag{13}$$

comes about. First, consider $\rho$ to be a pure local product state. The information in the total system is then $L \log_2 d$, where $L$ is the number of sites and $d$ is the local Hilbert space dimension. Since all single site density matrices are pure, the information on each site is $i_n^0 = \log_2(d)$, and since there are $L$ sites these terms add up to $L \log_2(d)$. All other terms are zero, since there is no shared information between sites. As a second example consider the dimerized state of spin-1/2's where every other pair of adjacent spins is in a singlet state. Then all single site density matrices are maximally mixed so all terms $i_n^0$ vanish. The pair of sites sharing a bond have a mutual information 2, and there are $L/2$ such pairs adding up to $L$. The pair of adjacent sites not sharing a bond are maximally mixed and their corresponding mutual information is zero. There is no correlations between nonadjacent sites, so all values with higher $l$ vanish, and the left- and right hand side of Eq. (13) again coincide.

The general case (12) is proved by induction. That the sum (12) holds for $l = 0$ in $I(\mathcal{C}_n^l)$ follows directly from the expression (5) for $i_n^0$. Assume that (12) holds for $l' < l$. Using the property $S_n^l = \{l, n\} \cup S_{n-\frac{1}{2}}^{l-1} \cup S_{n+\frac{1}{2}}^{l-1}$ and $S_{n-\frac{1}{2}}^{l-1} \cap S_{n+\frac{1}{2}}^{l-1} = S_n^{l-2}$ we have

$$\sum_{(l', n') \in S_n^l} i_{n'}^{l'} = i_n^l + \overbrace{\sum_{\substack{(l', n') \\ \in S_{n-\frac{1}{2}}^{l-1}}} i_{n'}^{l'}}^{= I(\mathcal{C}_{n-1/2}^{l-1})} + \overbrace{\sum_{\substack{(l', n') \\ \in S_{n+\frac{1}{2}}^{l-1}}} i_{n'}^{l'}}^{= I(\mathcal{C}_{n+1/2}^{l-1})} - \overbrace{\sum_{\substack{(l', n') \\ \in S_n^{l-2}}} i_{n'}^{l'}}^{= I(\mathcal{C}_n^{l-2})}, \tag{14}$$

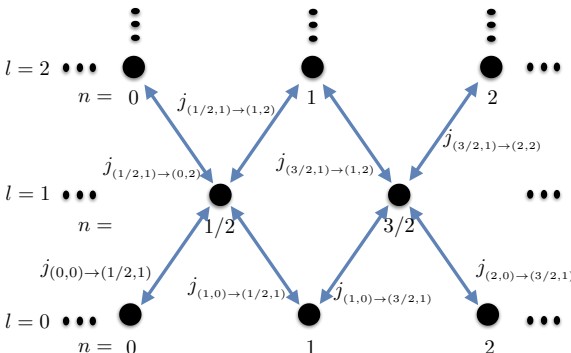

Figure 3: With a nearest neighbor Hamiltonian the information current $j_{(n',l')\to(n,l)}$ only connects nearest-neighbors in the information lattice.

where the equalities above the expressions follow from the induction assumption. From the definition of $i_n^l$ in Eq. (9), and the definition of $I(A;B)$ in Eq. (8), we get the correct sum (12) also for $l$, completing the proof.

## 2.3 Information currents

For a local Hamiltonian, the conservation of the total information is not just a global conservation law; in analogy to how currents are defined given a locally conserved operator, the local structure of the information lattice gives rise to well-defined local information currents, see Fig. 3. Consider a operator $\hat{Q} = \sum_n \hat{q}_n$, where $\hat{q}_n$ acts on site $n$, that commutes with a 1D nearest neighbor Hamiltonian

$$H = \sum_n h_{\{n,n+1\}}, \tag{15}$$

where $\{n,n'\}$ denotes an unordered pair of sites and $h_{\{n,n'\}}$ acts only on the sites $n, n'$. The time-derivative of the density matrix $\rho$ is decomposed into terms each stemming from a term in the Hamiltonian,

$$\dot{\rho} = \sum_n \overbrace{i[\rho, h_{\{n,n+1\}}]}^{\delta_{\{n,n+1\}}}. \tag{16}$$

Therefore, the conserved charge at each site $\dot{q}_n = \frac{d}{dt}\text{Tr}(\hat{q}_n \rho)$ is in turn decomposed into local terms each stemming from a single $\delta_{\{n,n+1\}}$,

$$\dot{q}_n = \alpha^n_{\{n-1,n\}} + \alpha^n_{\{n,n+1\}}. \tag{17}$$

Each $\delta_{\{n,n+1\}}$ contributes to the time-derivative of the conserved charge on two sites, $n$ and $n + 1$, and the assumption $[H, \hat{Q}] = 0$ implies that the contribution is equal up to a sign: $\alpha^n_{\{n,n'\}} = -\alpha^{n'}_{\{n,n'\}}$. Therefore the decomposition of $\dot{\rho}$ into $\sum_n \delta_{\{n,n+1\}}$ gives rise to a well-defined definition of the flow of charge from site $n + 1$ to site $n$, $j_{n+1\to n}$, given by

$$j_{n+1\to n} = \alpha^n_{\{n,n+1\}}. \tag{18}$$

Turning to the information lattice, we let $\alpha^{(n,l)}_{\{n,n+1\}}$ denote the term in $i_n^l$ stemming from $\delta_{\{n,n+1\}}$ in the decomposition of $\dot{\rho}$ (16),

$$i_n^l = \alpha^{(n,l)}_{\{n-l/2-1,n-l/2\}} + \alpha^{(n,l)}_{\{n+l/2,n+l/2+1\}} + \alpha^{(n,l)}_{\{n-l/2,n-l/2+1\}} + \alpha^{(n,l)}_{\{n+l/2-1,n+l/2\}}. \tag{19}$$

Analogously to the usual conserved charge, each term is only present in the decomposition of the time-derivative of a single other information lattice value, but then with reversed sign, e.g.,

$$\alpha^{(n,l)}_{\{n-l/2-1,n-l/2\}} = -\alpha^{(n-1/2,l+1)}_{\{n-l/2-1,n-l/2\}}. \tag{20}$$

So, we have a well-defined notion of the local currents,

$$j_{(n+1/2,l-1)\to(n,l)} = \alpha^{(n,l)}_{\{n-l/2,n-l/2+1\}}, \qquad j_{(n+1/2,l-1)\to(n,l)} = \alpha^{(n,l)}_{\{n+l/2-1,n+l/2\}}, \tag{21}$$

$$j_{(n+1/2,l+1)\to(n,l)} = \alpha^{(n,l)}_{\{n+l/2,n+l/2+1\}}, \qquad j_{(n-1/2,l+1)\to(n,l)} = \alpha^{(n,l)}_{\{n-l/2-1,n-l/2\}}. \tag{22}$$

At first sight it might seem odd that the left most term $h_{\{n-l/2,n-l/2+1\}}$ is responsible for the current from the right line-segment subset, and not the other way around. This is however not as unintuitive as it might seem: the term which can get correlations between the $l$ right most sites in $\mathcal{C}^l_n$ to spread and become a correlation involving all $l+1$ sites is precisely $h_{\{n-l/2,n-l/2+1\}}$.

The given expressions for the currents are in terms of derivatives of $i^l_n$, which we now want to write in closed-form expressions. The gradient $\nabla f$ of smooth scalar functions $f$ on the space of Hermitian matrices is the matrix satisfying

$$\mathrm{Tr}(\nabla f[\rho]\Delta\rho) = \lim_{\epsilon\to 0}\frac{f[\rho+\epsilon\Delta\rho]-f[\rho]}{\epsilon}, \tag{23}$$

for any Hermitian matrix $\Delta\rho$. From this definition the gradient $\nabla S[\rho]$ of the von Neumann entropy is

$$\nabla S[\rho] = -\log_2(\rho) - \mathbb{1}/\ln(2). \tag{24}$$

Since $i^l_n$ is a sum of von Neumann entropies, see Eqs. (9) and (8), we can use this result to get an expression for the gradient of $i^l_n$,

$$\nabla i^l_n = \log_2(\rho_{\mathcal{C}^l_n}) + \log_2(\rho_{\mathcal{C}^{l-2}_n}) - \log_2(\rho_{\mathcal{C}^{l-1}_{n-1/2}}) - \log_2(\rho_{\mathcal{C}^{l-1}_{n+1/2}}). \tag{25}$$

The coefficient $\alpha^{(n,l)}_{\{n-l/2,n-l/2+1\}}$ is of the form of the right side of the definition of the gradient (23) with $\Delta\rho = \delta_{\{n-l/2,n-l/2+1\}} = i[\rho, h_{\{n-l/2,n-l/2+1\}}]$ and $f = i^l_n$. So,

$$\alpha^{(n,l)}_{\{n-l/2,n-l/2+1\}} = i\,\mathrm{Tr}\Big(\nabla i^l_n\,[\rho_{\mathcal{C}^l_n}, h_{n-l/2}]\Big), \tag{26}$$

where we introduced the short-hand notation $h_n \equiv h_{\{n,n+1\}}$. Inserting the expression (25) for the gradient $\nabla i^l_n$ we thus have a closed form expression for the current $j_{(n+1/2,l-1)\to(n,l)}$ involving only the reduced density matrices. Doing the analogous rewriting for the three other currents we get closed form expressions for all currents,

$$j_{(n+1/2,l-1)\to(n,l)} = i\,\mathrm{Tr}\Big(\nabla i^l_n\,[\rho_{\mathcal{C}^l_n}, h_{n-l/2}]\Big), \tag{27}$$

$$j_{(n-1/2,l-1)\to(n,l)} = i\,\mathrm{Tr}\Big(\nabla i^l_n[\rho_{\mathcal{C}^l_n}, h_{n+l/2-1}]\Big), \tag{28}$$

$$j_{(n+1/2,l+1)\to(n,l)} = i\,\mathrm{Tr}\Big(\nabla i^l_n[\rho_{\mathcal{C}^{l+1}_{n+1/2}}, h_{n+l/2}]\Big), \tag{29}$$

$$j_{(n-1/2,l+1)\to(n,l)} = i\,\mathrm{Tr}\Big(\nabla i^l_n[\rho_{\mathcal{C}^{l+1}_{n-1/2}}, h_{n-l/2-1}]\Big). \tag{30}$$

Finally, for later reference, we also introduce the notation $\mathcal{J}_{l\to l+1}$ for the total current from scale $l$ to $l+1$,

$$\mathcal{J}_{l\to l+1} = \sum_{\text{all }n} j_{(n,l)\to(n-1/2,l+1)} + j_{(n,l)\to(n+1/2,l+1)}, \tag{31}$$

and $j_{l \to l+1}$ (without any position index) for the total current per site. The total current is a $1D$ current which means it also can be defined directly from the continuity equation,

$$\mathcal{J}_{l \to l+1} = -\frac{d}{dt}\sum_{l'=0}^{l}\mathcal{I}^{l'}, \tag{32}$$

where

$$\mathcal{I}^{l} = \sum_{\text{all } n} i_{n}^{l}. \tag{33}$$

## 3 Thermalization dynamics

We are now in position to discuss the general properties of thermalization dynamics from the perspective of the information lattice. To this end, we study the evolution of $i_{n}^{l}$ on the information lattice in two different situations: first from a homogenous initial state and then from an initial state which is homogenous except at one point where there is a perturbation. In both cases we employ the nonintegrable transverse- and longitudinal-field quantum Ising Hamiltonian,

$$H = \sum_{n} h_{n}, \quad h_{n} = J s_{n}^{z} s_{n+1}^{z} + \frac{1}{2}\left(h_{L}(s_{n}^{z} + s_{n+1}^{z}) + h_{T}(s_{n}^{x} + s_{n+1}^{x})\right), \tag{34}$$

where the operators $s_{n}^{x}$ and $s_{n}^{z}$ are spin-half (with eigenvalues $\pm 1/2$) operators on site $n$. The specific values of the Ising parameters are not very important; for easy comparison we take them as in Ref. [10], $h_{L} = 0.25J$ and $h_{T} = -0.525J$.

For the first example we consider a quench from the initial state,

$$\rho(t=0) = \bigotimes_{n}\rho_{n}, \quad \rho_{n} = \frac{2}{3}|\uparrow\rangle\langle\uparrow| + \frac{1}{3}|\downarrow\rangle\langle\downarrow|, \tag{35}$$

time-evolved with the Hamiltonian (34). The information in the initial state is purely local and, as shown in Fig. 4, remains so at short times. As can be seen in Fig. 4a, later at $t = 10J^{-1}$ and $t = 14J^{-1}$, the information has split into two main parts: one part travels to larger and larger scales at the Lieb-Robinson speed [20] (reminiscent of the entanglement tsunami in holographic systems [21]), and the other remains stationary and purely local at small scales. Note also how the curves, in Fig. 4a, for $l \le 2$, at $Jt = 10$ and $Jt = 14$ are indistinguishable. This local part corresponds to the local density matrices of the thermalized infinite-time state.

In Fig. 4b, slightly after $t = 6J^{-1}$, the splitting of the information is visible: a gap opens up forming two separate information bumps. If the information at scale $l$ is zero, it means that local density matrices at scale $l$ can be reconstructed from the density matrices at scale $l - 1$. In turn, this means that the $(l - 1)$-local density matrices can be time-evolved without any knowledge of longer-range correlations; the local degrees of freedom have decoupled from the rest. It is, however, not required that the information at a scale completely vanishes for decoupling to occur. In fact, the information current, depicted in Fig. 4, also vanishes at the smallest scales when the information wave-packet is well separated. This vanishing of information current is sufficient for decoupling. For statistical reasons, information generically flows from small scales to large. When the information current from $l$ to $l + 1$ vanishes one therefore generically expects that, up to local constraints, the information in the $l$ smallest scales is minimal. In this case we can reconstruct the $(l + 1)$-local density matrices from the $l$-local density matrices via the state with minimal information given the $l$-local density matrices:

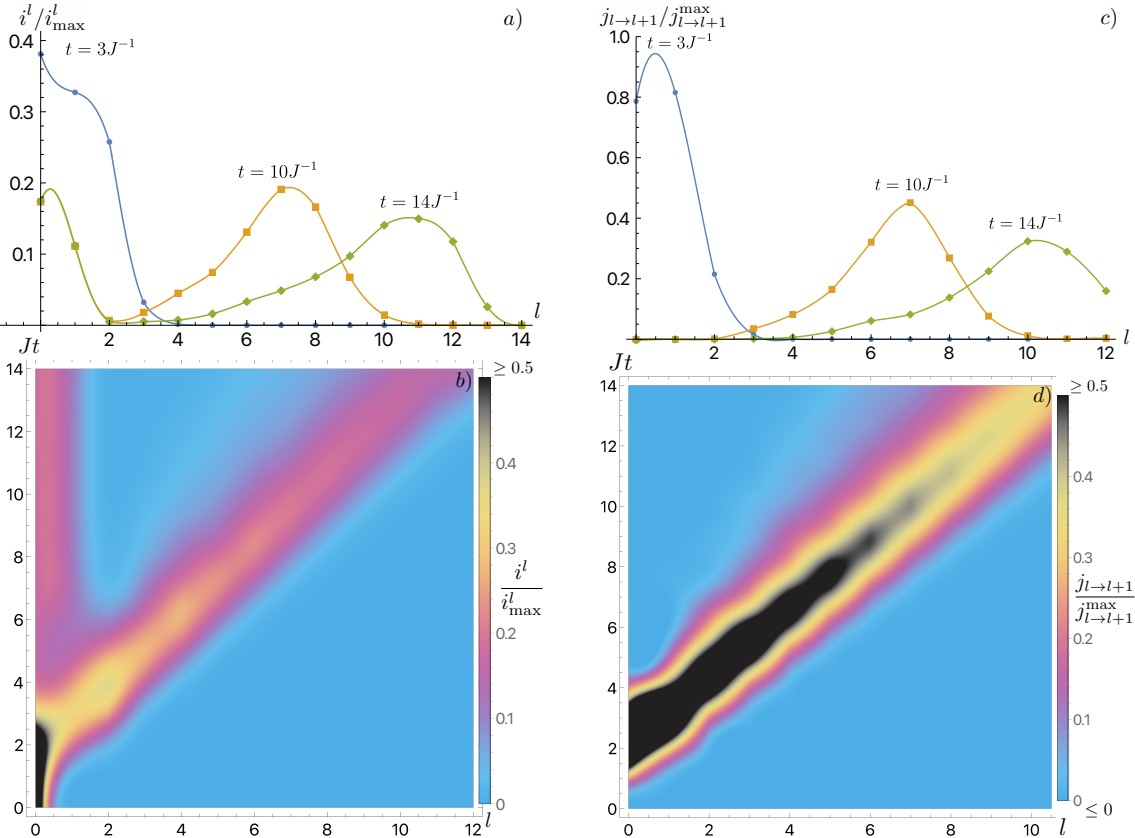

Figure 4: Time-evolution of information in the transverse- and longitudinal-field Ising model with the infinite product state with $\rho_n = \frac{2}{3}| \uparrow \rangle\langle \uparrow | + \frac{1}{3}| \downarrow \rangle\langle \downarrow |$ as initial state. **a)** The information lattice values $i^l$ as a function of $l$ at three different times in units of the maximum information lattice value $i^l_{\max} = 5/3 - \log_2(3)$ ($n$ is suppressed due to translation invariance). At $t = 3J^{-1}$ (blue dots), nearly all the information remains local at scales $l < 4$. At $t = 10J^{-1}$ (orange squares) two peaks have formed with almost zero information in between (note that up to $l = 2$ the green curve lies directly on top of the orange curve obscuring its view). As the system continues to evolve at $t = 14J^{-1}$ (green diamonds) the information for $l \leq 2$ has essentially stabilized to its infinite-time value while the peak at long range travels to larger and larger scales. **b)** The information lattice values $i^l$ for a time continuum. Notice the gap between the information localized at the smallest scales and the peak traveling to larger and larger scales which is beginning to form slightly after $t = 6J^{-1}$. **c)** The total information-current per site $j_{l \to l+1}$ in units of the maximum current $j^{\max}_{l \to l+1} \approx 0.021$. **d)** The total information-current for a time continuum. (In all plots the values at non-integer $l$, obtained by a third order spline interpolation, are added as a guide to the eye.)

the $l$-local Gibbs state, see App. E. The reconstructed $(l + 1)$-local density matrices then give the time-derivative of the $l$-local density matrices, making the time-evolution of the $l$-local density matrices closed.

It is important to note that care must be taken in choosing $l$, when approximating a state with an $l$-local Gibbs state. In the example illustrated in Fig. 4, we get at $t = 10J^{-1}$ an accurate approximation of the derivative of the 3-local density matrices using a 3-local Gibbs state defined by the 3-local density matrices. However, if we instead use, e.g., a 7-local Gibbs state defined by the 7-local density matrices, we do not get an accurate approximation of

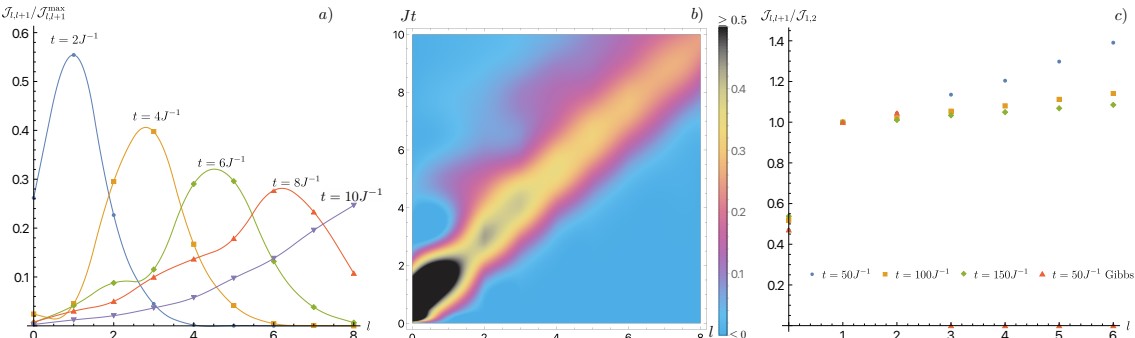

Figure 5: Time-evolution of information in the transverse- and longitudinal-field quantum Ising Hamiltonian starting from the initial state (36) that is the product state of maximally mixed states except at a single site where the spin points in the positive $x$-direction. **a)** The total information-current in units of the maximum total information current $\mathcal{J}_{l \to l+1}^{\max} \approx 0.45$ at several different times (before the dynamics is dominated by diffusion). **b)** The total information-current in units of the maximum total information current $\mathcal{J}_{l \to l+1}^{\max} \approx 0.45$ for a continuum of times. **c)** The information-currents at late times in units of $\mathcal{J}_{1 \to 2}$, which equals $\approx 15 \times 10^{-5}$ for $t = 50$, $\approx 4.2 \times 10^{-5}$ for $t = 100$ and $\approx 2.1 \times 10^{-5}$ for $t = 150$. As a comparison, the current in a 3-local Gibbs state, with the same local density marices, at $t = 50J^{-1}$, is shown. The Gibbs state underestimates the current by several orders of magnitude; For example, at $t = 50J^{-1}$ the $\mathcal{J}_{3,4}$ current is underestimated by a factor of $1.3 \times 10^4$, it continues to decay, and $\mathcal{J}_{7,8}$ is underestimated by a factor of $1.4 \times 10^{12}$. In **a)** and **b)** the values at non-integer $l$, given by third order spline interpolation, are added as a guide to the eye.

the time-derivative of the 7-local density matrices. The reason is that such an $l$-local Gibbs state would severely underestimate the information currents at scales $> 7$. The accumulation of information at scale 7 will lead to an erroneous flow back to smaller scales and spoil the dynamics of the local density matrices. The same would be true if we tried to approximate the derivative using a matrix product state (MPS) or a matrix product density operators (MPDO) (or any other technique aimed at approximating equilibrium type states): using the minimal bond-dimension MPS or MPDO which captures the 7-local density matrices will generically severely underestimate the information current on larger scales.

In the example of Fig. 4, the local density matrices are static after the local degrees of freedom have decoupled and the time-evolution to infinite time is captured by just time-evolving until that decoupling time. However, decoupling of local degrees of freedom does not necessarily imply that the local density matrices are static: Consider as an example a state which thermalizes into local excitations that then bounce around like billiard balls. The dynamics continues forever and the full dynamics can not be captured by time-evolving until some finite time. At the same time, the information that left the small scales before reaching local equilibrium will continue to travel to larger and larger scales such that the resources for time-evolving the full state grow exponentially with time.

A perfect splitting of information into two bumps is not generic. An inhomogeneous distribution of a locally conserved quantity has to spread diffusively before the last part of the information in the small scales can leave. Therefore, such an initial distribution leads to a slow trickle, with a magnitude only decaying algebraically with time, of information from small to large scales. However, it is not only, e.g., $H$ itself which is conserved; products, e.g., $H^2$, $H^3$, etc., are also conserved operators. Generically the corresponding correlation functions, e.g., $\langle h_n h_n' \rangle$ approach their equilibrium value polynomially [22, 23]. However, the operators be-

come less local as you consider larger products; thus, the impact on local density matrices becomes smaller and smaller. In the example here, we both start from a product state, and the eventual equilibrium state also has a minimal correlation length, which implies that the prefactor of the algebraically decaying correction to the local density matrices is minuscule. Here, we see the almost perfect gap between the bump of information going to infinity and the one staying at local scales. (A closer inspection shows a minor correction to the information current at intermediate scales, which decays slowly).

In our next example we consider a time-evolution where decoupling of the local degrees of freedom by $l$-local Gibbs states does not become a good approximation. We consider the time-evolution of a state which initially has an inhomogeneous distribution of a conserved charge and eventually relaxes to an infinite temperature state. This inhomogeneous distribution diffuses and smoothens over time, leading to a slow trickle of information out of the smallest scales, meaning that the information current will at no time and scale become small compared with the information at the smallest scales. We use the same Hamiltonian as before, on an infinite one-dimensional chain, with initial state the product state of maximally mixed states on all but one site (as in Ref. [10]):

$$\rho(t=0) = \cdots \otimes I_2 \otimes I_2 \otimes |\!\uparrow_x\rangle\langle\uparrow_x\!| \otimes I_2 \otimes I_2 \otimes \cdots, \tag{36}$$

where $I_2$ is half the identity matrix. The conserved charge in this case is energy, and there is an excess energy around the site where a spin initially points up. This energy will spread out, leading to a gradual decrease of the local density marices. This can be seen in Fig. 5 that shows the time-evolution of the information current. As in the first example in Fig. 4, there is an information-current wave packet that travels to larger and larger scales. Now, however, it leaves behind a substantial tail extending to small scales, and the information current never vanishes. Eventually, everything but the diffusive dynamics is damped out. The smallest scales carry information about the energy and there is a constant information flow from the smallest scales that slowly decreases over time (since diffusion slows down as the energy distribution become increasingly smooth). Since there is nothing that constrains this information we expect it to flow with a constant speed toward infinite scales. This means that there is no sharp scale $l$ at which the total information current, $\mathcal{J}_{l\to l+1}$, becomes much smaller than on other scales. Instead, $\mathcal{J}_{l\to l+1}$ slowly increases with $l$, for $l$ small compared to the scale that the main information wave packet, traveling to infinity, has reached.

An intuitive picture of the increase of the information current with $l$ is available if we assume that information leaving the smallest scales travels only in one direction, namely to larger and larger scales. Looking at the information current at larger $l$ is then akin to looking back in time, as it carries the information which left the smallest scales in the past. This behavior can be seen in Fig. 5c, where the information current is slowly increasing as a function of $l$, with a slope that decreases with time. The only exception is $\mathcal{J}_{0\to 1}$ which reflects dynamics on a scale smaller than the range of the Hamiltonian, where the above argument is not valid.

In this case there is no scale at which an $l$-local Gibbs state provides a good approximation. As an example, in Fig. 5c, we also show the information current for a 3-local Gibbs state, which severely underestimates the current at scale $l$ and larger. The same is true for an MPS or MPDOs, even if they are chosen to correctly capture the $l$-local density matrices they will severely underestimate the information current on scales $\sim \log_d \chi$. In the next section we will discuss an idea for how to capture this situation.

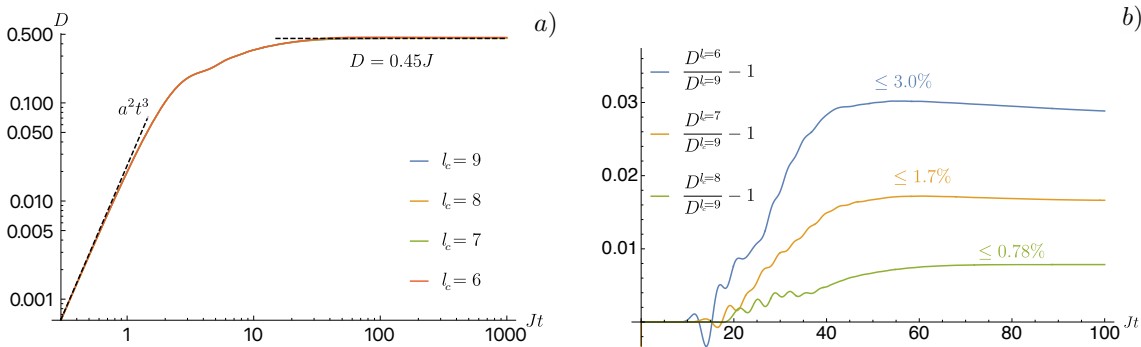

Figure 6: **a)** The diffusion coefficient as a function of time $t$ and cut-off scale $l_c$ (at this scale the curves are on top of each other), starting from the initial state defined in Eq. (41). After a brief initial ballistic evolution (for a duration of order $\sim J^{-1}$), we observe a significantly longer crossover period before normal diffusion is reached, with a constant diffusion coefficient. **b)** The relative error of the three truncation variables (the error is defined by comparing to the largest truncation value, $l_c = 9$.)
.

## 4 Time-evolving local density matrices

In this section, we build on the intuition gained from our study of information flow during thermalising dynamics to develop algorithms to time-evolve the $l$-local density matrices. We first introduce the general framework for such algorithms, before discussing a concrete algorithm.

As before, we take $\Omega^l$ and $\Omega^{l+1}$ to be the $l$ and $(l+1)$-local density matrices of a given quantum state. For a Hamiltonian with nearest-neighbor couplings, the time-derivative $\dot{\Omega}^l$ is a linear map $\boldsymbol{\Phi}$ of $\Omega^{l+1}$, i.e.,

$$\dot{\Omega}^l = \boldsymbol{\Phi}(\Omega^{l+1}), \tag{37}$$

as follows directly from the properties of the partial trace and the Heisenberg equation of motion. As a concrete example, consider a $1D$ system and the time-derivative of an element in $\rho_{[n,n+l]} \in \Omega^l$. If the Hamiltonian $H$ only has nearest-neighbor terms then the time-derivative $\rho_{[n,n+l]}$ can be obtained from elements exclusively in $\Omega^{l+1}$:

$$i\dot{\rho}_{\mathcal{C}_n^l} = \underset{(\mathcal{C}_n^l)^c}{\mathrm{Tr}} [H, \rho] = \sum_{m=n-l/2}^{n+l/2-1} [h_{\{m,m+1\}}, \rho_{\mathcal{C}_n^l}]$$
$$+ \mathbf{T}_L[h_{\{n-1,n\}}, \rho_{\mathcal{C}_{n-1/2}^{l+1}}] + \mathbf{T}_R[h_{\{n+l,n+l+1\}}, \rho_{\mathcal{C}_{n+1/2}^{l+1}}], \tag{38}$$

where the operator $\mathbf{T}_L$ ($\mathbf{T}_R$) is the trace operator tracing out the leftmost (rightmost) site of any operator on a line segment, e.g.,

$$\mathbf{T}_L \rho_{[n,n+l]} = \underset{n}{\mathrm{Tr}}\, \rho_{[n,n+l]}. \tag{39}$$

We introduce a cut-off $l_c$ in the locality of the information by approximating $\boldsymbol{\Phi}(\Omega^{l+1})$ by a compatible function $\boldsymbol{\Psi}$ of $\Omega^l$ only, such that

$$\dot{\Omega}^{l_c} \approx \boldsymbol{\Psi}(\Omega^{l_c}). \tag{40}$$

Compatible means that there exists some local density matrices $\tilde{\Omega}^{l_c+1}$ such that

$$\boldsymbol{\Psi}(\Omega^{l_c}) = \boldsymbol{\Phi}(\tilde{\Omega}^{l_c+1}), \tag{41}$$

with

$$\mathbf{T}_{l_c+1 \to l_c} \tilde{\Omega}^{l_c+1} = \Omega^{l_c}, \tag{42}$$

where $\mathbf{T}_{l+1 \to l}$ is the trace operator which is a linear map from the $(l+1)$-local density matrices to the $l$-local density matrices; in $1d$ it takes the form,

$$\mathbf{T}_{l+1 \to l}\{\rho_{\mathcal{C}_n^l}\}_{\text{all } n} = \{\mathbf{T}_R \rho_{\mathcal{C}_n^l}\}_{\text{all } n} \cup \{\mathbf{T}_L \rho_{\mathcal{C}_n^l}\}_{\text{rightmost } \mathcal{C}_n^l}. \tag{43}$$

The compatibility requirement means that at each time step errors are only introduced on scales larger than $l_c$. One consequence is that any $l$-local conserved quantity, with $l \le l_c$, is left invariant, i.e., the expectation value of any operator $\mathcal{O}$ of the form

$$\mathcal{O} = \sum_n \omega_n^l, \quad \omega_n^l \text{ acts on } \mathcal{C}_n^l, \tag{44}$$

such that $[\mathcal{O}, H] = 0$, is conserved by the time-evolution.

We want to capture dynamics in which the information not constrained to stay at small scales can be assumed to flow by statistical drift to larger and larger scales, and therefore never comes back to affect the local degrees of freedom. Any $\Psi$ which does not obstruct this flow can then be used to predict the dynamics of the local degrees of freedom: for large enough $l_c$, the global flow of information guarantees that the algorithm accurately captures the dynamics of the $l'$-local density matrices, for small $l'$. The question is then how to find a $\Psi$ which does not obstruct the information flow.

Using Petz recovery maps, if the information in layer $l_c + 1$ is small, we can extend the density matrices from scale $l_c$ to scale $l_c + 1$ with a controlled error given by the bound (11). We use this method in the first simulation in Fig. 4, and at early times also in the other simulation, we define

$$\Psi(\Omega^{l_c}) = (\Phi \circ \mathcal{M}_{\text{Petz}})(\Omega^{l_c}), \tag{45}$$

where $\mathcal{M}_{\text{Petz}}$ is defined by first using a Petz map to extend the density matrices on scale $l_c$ to density matrices on scale $(l_c + 1)$ and then projecting this set of density matrices onto the space fulfilling the consistency condition (42) (see App. C for details). We can thus time-evolve the local density matrices with a known bound on how far the density matrices are from the true density matrices which one would have gotten by time-evolving the entire state according to the Schrödinger equation. If information is initially local, i.e., $i_n^l \approx 0$ for $l > l'$ and $l' < l_c$ then it will take time $T \sim (l_c - l')/v$, where $v$ is the Lieb-Robinson speed, before any information reaches scale $l_c$, and we can thus always initially time-evolve until time $\sim T$ with a small bound on the error. If there during time $T$ is some scale $\tilde{l} < l_c$ where an information gap opens, then we can, using the above choices for $\Psi$, continue to time-evolve the local density matrices accurately to arbitrarily late times if we use the cut-off $l_c = \tilde{l}$. So in that case, one can time-evolve local density matrices to arbitrary late times without needing resources growing exponentially with time [3].

The challenge that remains is to time-evolve the $l_c$-local density matrices if no such gap opens. At a first glance it might seem like a good idea to define $\Psi$ by removing the information

---

[3]In $1D$, one can consider an equivalent time-evolution algorithm based on MPSs. There are several MPS based techniques to accurately time-evolve states that start out with only local density matrices for a finite amount of time. For pure states one can use TEBD and for mixed states one can, e.g., use TEBD together with purification [24,25]. After local equilibrium has emerged one can, using the algorithm from Ref. [26], generate an MPDO with a given $l$-local density matrices. Then, since information stays local, the time-evolution can be continued to arbitrary times without the bond-dimension growing exponentially. (Since generalized Gibbs states generically are MPDOs with finite bond dimension [27] it is reasonable to assume that constructing an MPDO from a the $l$-local density matrices is a good approximation to the $l$-local Gibbs state given the $l$-local density matrices. In this case, time-evolving the MPDO will give an accurate prediction of the dynamics of the $l$-local density matrices [27]).

on scales larger than $l_c$. At every time step, such an algorithm discards all information at scales larger than $l_c$. However, while it does not create any erroneous information, it will in general underestimate the information flow leaving the $l_c$ smallest scales when applied to more generic situations, as shown in Fig. 5. Almost all information that should have disappeared to large scales, with the main wave packet, instead builds up at scale $l_c$. Since most of the information typically disappears to infinity, the time-evolution sees an erroneous buildup of information, which can become much larger than the information in the degrees of freedom we are trying to capture.

To avoid this unphysical information buildup we construct an algorithm by assuming—from statistical arguments—that the precise correlations on intermediate scales are of no importance as long as they are responsible for carrying the information leaving smaller scales to infinity. We therefore approximate the currents $\{j_{(l_c,n)\to(l_c+1,n')}\}_{n,n'}$ as a function of the $l_c$-local density matrices. In general, one expects that in addition to the general flow to larger and larger scales there is a diffusion of information so that information flows from points in the information lattice with more information, to points with less information. For the sake of simplicity we assume that it suffices to correctly capture the total flow toward larger scales, that is to say to approximate the total current $\mathcal{J}_{l_c\to l_c+1}$ instead of the entire set $\{j_{(l_c,n)\to(l_c+1,n')}\}_{n,n'}$; extensions to local flows are in principle possible. A more precise treatment of the information diffusion is kept for later work.

At short times, no information leaves the $l_c$ smallest scales, and the state is an $l_c$-local Gibbs state. As can be seen in Fig. 5, as time progresses, the total current becomes roughly constant as a function of $l$

$$\mathcal{J}_{l\to l+1} \approx \mathcal{J}_{l-1\to l}. \tag{46}$$

These two extremal situations can be connected through the following insight: If $\mathcal{I}_l$, the total information on scale $l$, is large, the flow leaving scales $l$ should also be large. We model this by assuming that the current $\mathcal{J}_{l\to l+1}$ is proportional to $\mathcal{I}_l$ which gives us the approximation

$$\mathcal{J}_{l_c\to l_c+1} = \frac{\mathcal{I}_{l_c}}{\mathcal{I}_{l_c-1}} \mathcal{J}_{l_c-1\to l_c}. \tag{47}$$

While being a somewhat rough approximation, it is also (partially) self-correcting: if we underestimate the current $\mathcal{J}_{l_c\to l_c+1}$ then $\mathcal{I}_{l_c}$ will grow and therefore the current will also grow.

Specifying the current does not suffice to specify $\mathbf{\Psi}$ and thus the time derivative $\dot{\Omega}^{l_c}$. The remaining degrees of freedom, though assumed to be globally unimportant, cannot be chosen completely arbitrarily. The self-correcting property of the current condition (47) guarantees a certain average current flow. However, certain choices of the remaining degrees of freedom could still result in an oscillating information with a large amplitude which we would expect leads to a slow convergence as a function of $l_c$. To avoid this situation, we try to make $I_{\text{tot}}^{l_c} = \sum_{l'=0}^{l_c} I^{l'}$ smooth. More precisely, we use the second order Taylor expansion of $I_{\text{tot}}^l$ as a measure. Let $\chi$ be a possible choice for the time-derivative of $\Omega^{l_c}$:

$$\chi \in \mathbf{\Phi}(\mathcal{C}_{\Omega^{l_c}}^{l_c+1}), \tag{48}$$

where $\mathcal{C}_{\Omega^l}^{l+1}$ denote the space of $(l+1)$-local density matrices compatible with $\Omega^l$, i.e.,

$$\tilde{\Omega}^{l+1} \in \mathcal{C}_{\Omega^l}^{l+1} \quad \Longleftrightarrow \quad \mathbf{T}_{l+1\to l}\tilde{\Omega}^{l+1} = \Omega^l. \tag{49}$$

If we change $\Omega^{l_c}$ in the direction $\chi$, $I_{\text{tot}}^{l_c}$ changes as

$$I_{\text{tot}}^{l_c}(\Omega^{l_c} + \epsilon\chi) = I_{\text{tot}}^{l_c}(\Omega^{l_c}) - \epsilon\mathcal{J}_{l_c\to l_c+1}(\chi) + \frac{\epsilon^2}{2}b_{\Omega^{l_c}}(\chi,\chi) + \mathcal{O}(\epsilon^3). \tag{50}$$

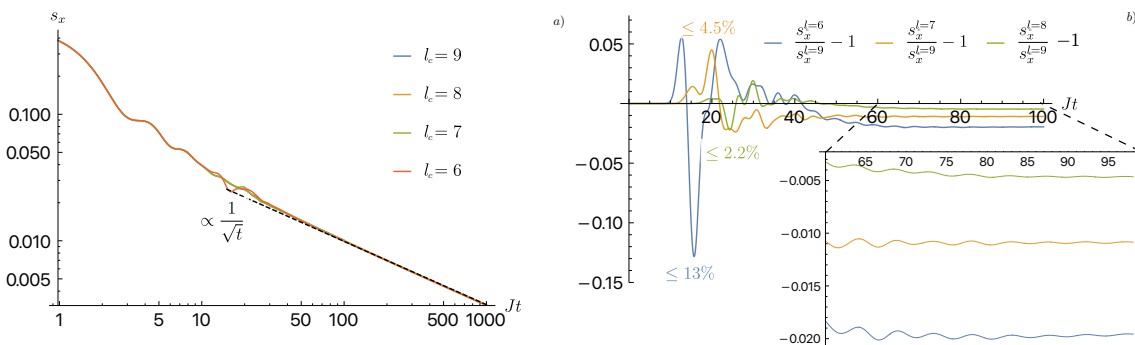

Figure 7: **a)** Expectation value of the spin on the central site $n_0$ as a function of time for the initial state definied in Eq. (36) and for truncation values $l_c$ from 6 to 9. At late times, it follows the $1/\sqrt{t}$ behavior expected from conventional diffusion. **b)** The relative error on $\langle s_x \rangle$, using the largest truncation value $l_c = 9$ as reference value. We indicate on the graph the largest relative error for each truncation value. Although the maximum error is larger than for the diffusion constant, the two largest truncation values agree everywhere on the two leading digits. Also the error stabilizes to roughly, but somewhat smaller, value than for the diffusion constant.

The first order term is directly specified by the current condition (47). So, we choose $\chi \in \Phi(\mathcal{C}^{l_c+1}_{\Omega^{l_c}})$ to minimize the bilinear map, $b_{\Omega^{l_c}}(\chi, \chi)$, given that the current condition is fulfilled. The bilinear form $b_{\Omega_{l_c}}$ is positive definite, so we simply have to minimize it to get the map $\Psi$. However, doing the Taylor expansion to define $b_{\Omega_{l_c}}$ and the following minimization naively leads to a slow numerical algorithm. In App. B we show how it can be done efficiently by first doing the Taylor expansion and part of the minimization analytically before a numeric step.

## 5 Numerical Simulations

We now discuss the time-evolution of the local density matrices $\Omega^{l_c}$ with the initial state (36), using the information flow algorithm of last section, with $\Psi$ defined by the current condition (47) and minimizing the expansion of $I^{l_c}_{tot}$ (50). At early times when the flow of information from scale $l_c$ to scale $l_c + 1$ is approximately zero, the analytical expression for $\Psi$ in the information flow algorithm is a good approximation of the exact time-derivative of the $l_c$-local density matrices. However, at the same time the denominator in the current condition (47) is small leading to potential numerical instability, which we fix by first time-evolving using the Petz recovery map (45).

The information-flow algorithm uses $l_c$ as a truncation variable. For $l_c \to +\infty$, it trivially reproduces the exact time-evolution at any finite time. At finite $l_c$, we estimate the error by the speed of convergence with $l_c$ of a few observables of interest. As the main estimator we use the relative error in the diffusion coefficient $D$, which characterizes the spreading of the energy distribution

$$D = \frac{1}{2} \frac{d}{dt} L^2(t), \tag{51}$$

where $L$ is the diffusion length:

$$L^2(t) = \frac{1}{\langle H \rangle} \sum_n (n + 1/2 - n_0)^2 \langle h_n \rangle_t. \tag{52}$$

Here $n_0$ denotes the lattice site of the spin initially in the state $|\uparrow_x\rangle$.

At short times, one generally expects a ballistic spread $L \sim vt$. However, our initial state is time-reversal invariant, enforcing $v = 0$. At short times, the diffusion length is therefore quadratic: $L \sim at^2$. (Since the initial state is a product state the acceleration can be calculated analytically: $a = h_T J/2\sqrt{3}$.) Later in the time-evolution, we instead expect no local reversibility, and thus random walk behavior $L \propto \sqrt{t}$. The diffusion coefficient then equals a constant—the diffusion constant. This behavior is seen in Fig. 6a. The dashed line at small times $\lesssim 1J^{-1}$ corresponds to cubically growing $D$, corresponding to the quadratically growing diffusion length. At late times $\gtrsim 50J^{-1}$ the diffusion coefficient is approximately the constant $D \approx 0.45J$ indicated by another dashed line. In between these limits there is a long crossover period $\sim 50J^{-1}$ with non-universal physics.

Our exact criterium for algorithmic convergence is that the maximum relative difference of the approximation of the diffusion coefficient with a truncation at scales $l_c - 1$ and a truncation at scale $l_c$ is smaller than 1%. In Fig. 6b we see that this requires a truncation variable $l_c = 9$ (this is also the highest truncation variable our optimized Mathematica code on a powerful desktop machine can handle). In the same figure we also see that, except for early times, the diffusion coefficient is always overestimated: the diffusion coefficient converges, as a function of $l_c$, from above.

Since we are time-evolving only sets of density matrices and not a quantum state, one might want to check if a global state exists for the system with the reduced density matrices we get from our algorithm. The general problem of verifying that a set of density matrices are compatible with a global state is QMA-complete[4] [28]. However, in the numerical examples we consider in this paper, the density matrices have at late times large smallest eigenvalues and one can verify that there exists $l$-local Gibbs states with short coherence length that have the $l$-local density matrices as reduced density matrices. (This does not mean that the global state is necessarily a Gibbs state, just that there exists a Gibbs state that is compatible). This can be verified with our algorithm in App. E. Nevertheless, we would like to stress that compatibility is not as crucial as one initially might think. It is not necessary to distinguish errors resulting in the $l_c$-local density matrices being incompatible with a global state and other errors. What matters is to estimate the total error made on the local density matrices. One could imagine working with density matrices incompatible with a global state but still only $\epsilon$ away from the correct local density matrices. In such a situation, these density matrices would only give an $\epsilon$ error to any local observable. Indeed, what matters for the local observables is not whether a global compatible state exists but the error in the local density matrices. As we discussed, in certain situations we do have a controlled bound on the error on the local density matrices. When we do not, we control the error with the convergence as a function of our truncation variable $l_c$.

Still, an important question for controlling the validity of our approach is whether the diffusion constant is an observable that is easier to capture accurately than others, since it is a purely universal property. In this particular quench most observables decay to zero exponentially fast and their relative error quickly becomes meaningless. However, the polarization $s_x$ at $n_0$ (the site of the initial perturbation) only decays algebraically. Having large $\langle s_x \rangle$ correlates with having a large energy. Even when most local information is gone, $\langle s_x \rangle$ is then simply tied to the energy diffusion, as shown in Fig. 7a. As seen in Fig. 7b the convergence is at first slower than for the diffusion coefficient, but still, at all times, agrees on the two leading digits for the two largest truncation values. However, as seen in the inset of Fig. 7b the late time convergence is roughly the same, or even slightly better, than for the diffusion coefficient.

Finally, we show in Fig. 8 that the information current also converges quickly with $l_c$. In

---

[4]Colloquially QMA-complete means that the problem is at least as hard as any other problem for which a quantum computer can verify that the solution is correct in polynomial time. So NP-complete is a subset of QMA-complete.

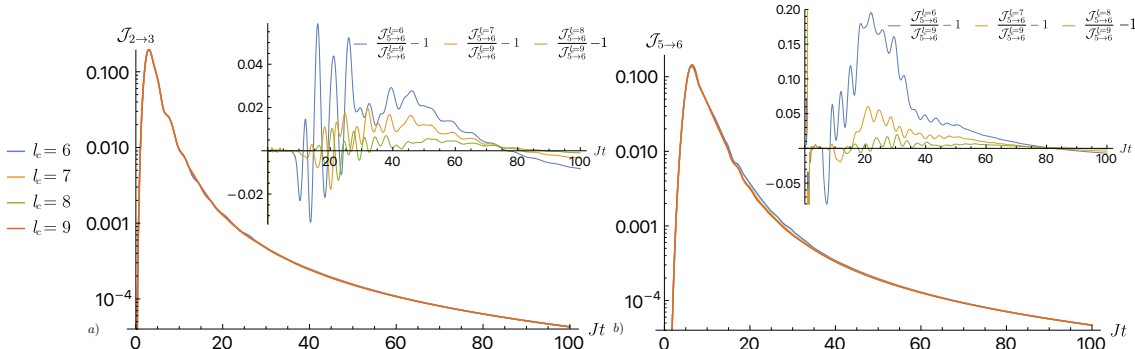

Figure 8: **a)** The information-current $\mathcal{J}_{2\to3}$ with the relative error as inset (using the largest truncation value $l_c = 9$ as reference). **b)** The equivalent plot $\mathcal{J}_{5\to6}$. The information-current $\mathcal{J}_{l\to l+1}$ shows good convergence as long as $l_c > l + 1$, and is on par with the convergence of the other observables we considered.

Fig. 8a it can be seen that the total information current $\mathcal{J}_{2\to3}$ initially converges faster than $\langle s_x \rangle$ and slower than the diffusion coefficient. At late times it shows roughly the same level of convergence. However, in Fig. 8b it can be seen that for the truncation value $l_c = 6$, $\mathcal{J}_{5\to6}$ has quite a substantial error of almost 20%. This is a generic behavior: for all truncation values, the $l_c$'th truncation value gives a bad approximation for the current $\mathcal{J}_{l_c-1\to l_c}$. The maximal relative error is 20%, 15% and 7% for $l_c = 6, 7$ and 8 respectively. This is simply a reflection of our approximation on the current condition in Eq. (47)—an error in the first unavoidably results in an error in the second.

It is worth noting that even the simple and imperfect current condition used here allowed for a high level of convergence in a long-time-evolution, in a nonintegrable model, at a remarkably low numerical cost. The difference between consecutive estimates of the diffusion constant with different $l_c$ decreases exponentially and reaches a level of less than 1%. This leads us to the conclusion that we could get a controlled estimate for the diffusion constant. Nevertheless, we expect that the current condition (47) is far from optimal and by improving it the convergence of the algorithm will be significantly faster.

## 6 Conclusion and Outlook

We have introduced the information lattice as a convenient and insightful way of capturing, in time and space, the flow of information during quantum time-evolution. This extends the physical lattice by an additional half-infinite dimension representing the scale on which the information in a quantum state is found. The information on a given scale with the corresponding information lattice coordinate $l$ is information that can not be found in any reduced density matrix of a size smaller than $l$. This allows for more fine-grained separation of entanglement, compared with, for example, matrix product states, which primarily focus on the largest entanglement eigenstates of a given bipartition. Since not all details of the entanglement are relevant for local observables, as much of the entanglement mainly serves to provide an effective bath to local degrees of freedom, such separation of entanglement scales offers new insights into quantum dynamics.

First, with the mixed transverse field Ising model as an example, we discussed dynamics where there is a finite time after which the flow of information vanishes at some scale. One can, in principle, capture such dynamics over an infinitely long time with finite resources with the methods we introduced. One could also use other methods, e.g., based on matrix product

states with limited bond dimensions; however, without the information lattice, it is hard to know how to implement them.

More generically, there is no finite time where the information flow vanishes and then all known algorithms with a controlled error break down. This situation is characterized by a slow flow of information to larger and larger scales. As most of the information that flows to larger scales never comes back to smaller scales and does not affect local observables, we can still obtain a long time-evolution of local observables. This requires keeping track of and resolving, not only the information (or entanglement) on small scales, but also, crucially, the flow of information at small scales.

With these insights we proposed a simple but highly efficient algorithm for time-evolution of quantum systems. Instead of time-evolving the full quantum state, we only time-evolve the local density matrices $\Omega^{l_c}$, which is the set of reduced density matrices of some size $l_c$. The exact time-evolution requires extending the scale $l_c$ at each time-step, but by simple assumptions about the structure of the information flow at the maximum scale we can close the time-evolution of the local density matrices $\Omega^{l_c}$—essentially by reconstructing $\Omega^{l_c+1}$ from $\Omega^{l_c}$ together with a physical assumption about the current flow out of scale $l_c$. The latter is essential: not keeping track of the information flow and only reconstructing $\Omega^{l_c+1}$ from $\Omega^{l_c}$ using a maximum entropy consideration, invariably results in unphysical backflow of information from large scales to small scales that can affect local observables. We have shown that this algorithm successfully captures diffusion at long times as well as the decay of local observables in the mixed transverse Ising model after a local quench from a thermal state with extra energy at one site.

While we have focussed our discussion on $1D$ models with nearest neighbor Hamiltonians, the essential concepts are readily generalized to both higher dimensions and longer range Hamiltonians. As our algorithm is based on local density matrices, it can likely also be generalized to include dissipation through local coupling to a bath. The algorithm does not rely on the presence of any symmetries, including translational invariance and can therefore by applied also to disordered systems. The complexity further only scales linearly with system size, assuming that a finite thermalization length scale emerges in the dynamics. Potential applications therefore include thermalization and many-body localization (or its absence) in higher dimensions, where no appropriate and efficient algorithms exists at the moment. We also expect that the information lattice will be useful in constructing analytical theories of information flow in thermalizing quantum systems. In particular, a more accurate and efficient modeling of the information flow at a given length scale will likely significantly improve the efficiency and accuracy of our algorithm.

## Acknowledgements

During the course of this project, we have had numerous discussions from which we have gained many insights. We especially want to acknowledge the insights of and discussions with Xiangyu Cao, David Aceituno, Daniel Parker, Sören Holst, and Ehud Altman.

**Funding information**   Thomas Klein Kvorning's (TKK) research is funded by the Wenner-Gren Foundations. Loïc Herviou was supported by the Roland Gustafsson's Foundation for Theoretical Physics and the Karl Engvers foundation. This work has also received funding from *Olle Engkvists Stiftelse* (SOEB) (grant No. 190-0381) and the European Research Council (ERC) under the European Union's Horizon 2020 research and innovation program (grant agreements No. 679722 and No. 101001902).

# A Notation and conventions

In this section we introduce notation and conventions which will be used in the following sections in the appendix.

In general we reserve greek letters (with superscript indicating scale) to denote sets of Hermitian operators each acting on a neighborhood, e.g.,

$$\psi^l = \{\psi^l_n\}_{\text{all } n}, \tag{53}$$

and $\psi^l_n$ is an operator on $\mathcal{C}^l_n$. As above we use a spatial subscript to denote elements of such sets. A greek letter with a superscript and a subscript, like $\psi^l_n$, should always be interpreted as the element of a set of Hermitian operators $\psi^l$ which acts in the neighborhood indicated by the sub- and superscripts. Using the same greek letter with different scale superscripts, i.e., $\psi^l$ and $\psi^{l-1}$, it should be understood that the sets are related via taking traces, in this case,

$$\psi^{l-1} = \mathbf{T}_{l \to l-1} \psi^l. \tag{54}$$

As before $\Omega^l$ is reserved to denote the $l$-local density matrices.

The sets of Hermitian operators form a real Hilbert space inherited from the real Hilbert space of Hermitian matrices, i.e., the vector addition and scalar multiplication are defined as

$$\psi^l + \phi^l = \{\psi^l_n + \phi^l_n\}_{\text{all } n}, \quad c\psi^l = \{c\psi^l_n\}_{\text{all } n}, \tag{55}$$

and the inner-product is defined by extending the trace inner-product to sets of Hermitian matrices as

$$\left\langle \{\psi^l_n\}_{\text{all } n} \,\middle|\, \{\phi^l_n\}_{\text{all } n} \right\rangle = \sum_n \text{Tr}(\psi^l_n \phi^l_n). \tag{56}$$

Maps between or in Hilbert spaces of Hermitian matrices or Hilbert spaces of sets of Hermitian matrices, are denoted by bold-face capital roman or greek letters as, e.g., $\mathbf{T}_{l \to l-1}$. We will refer to the adjoint of an operator with a superscript $T$ or with the word transpose since the Hilbert space is real. The transpose of an operator $\mathbf{O}$ from a set of Hermitian matrices of scale $l$ to a set of Hermitian matrices of scale $l'$ is the unique operator with the property

$$\langle \mathbf{O}\zeta^l | \tilde{\xi}^{l'} \rangle = \langle \zeta^l | \mathbf{O}^T \tilde{\xi}^{l'} \rangle, \tag{57}$$

for all $\zeta^l$ and $\tilde{\xi}^{l'}$. If the operator $\mathbf{O}$ is represented as a matrix the transpose amounts to the usual matrix transpose.

We denote the Moore-Penrose pseudoinverse (or just pseudoinverse) of an operator by a superscript +. The symbol $\mathbf{P}$ denotes orthogonal projectors, and if $\mathbf{O}$ is an operator then $\mathbf{P_O}$ denotes the orthogonal projector onto $\ker(\mathbf{O})$, the kernel of $\mathbf{O}$. It can be written in terms of the pseudoinverse as

$$\mathbf{P_O} = \mathbb{1} - \mathbf{O}^+ \mathbf{O}. \tag{58}$$

If $S$ is a linear space, then $\mathbf{P}_S$ denotes the orthogonal projection onto the space $S$.

We will use $\perp S$ to denote the orthogonal complement to $S$. The symbol $\mathbf{Q_O}$ denotes the orthogonal protector onto $\perp \ker \mathbf{O}$. In terms of the pseudoinverse

$$\mathbf{Q_O} = \mathbf{O}^+ \mathbf{O}. \tag{59}$$

Finally, $\mathbf{I_O}$ denotes the orthogonal projector onto $\text{im}(\mathbf{O})$, the image of $\mathbf{O}$. In terms of the pseudoinverse it can be written as

$$\mathbf{I_O} = \mathbf{O}\mathbf{O}^+. \tag{60}$$

# B   Details of the information-flow algorithm

In this section we explain how to construct the function for the derivative based on the current condition (47) and minimizing the second order of the information (50). We begin by introducing some notation and required mathematical objects.

## B.1   Preliminaries: linear operators

In this subsection we collect the expressions for the linear operators used in the rest of the section. First, the pseudo inverses of the left and right trace-operators, $\mathbf{T}_L$ and $\mathbf{T}_R$ defined in (39), act by tensor-multiplying $I_2$ to the left or the right,

$$\mathbf{T}_L^+ \psi_n^{l+1} = I_2 \otimes \psi_n^{l+1}, \tag{61}$$

$$\mathbf{T}_R^+ \psi_n^{l+1} = \psi_n^{l+1} \otimes I_2. \tag{62}$$

We will make use of the operator $\mathbf{P}_{\mathbf{T}_{l \to l-1}}$. To express it, note that $\psi^l \in \ker(\mathbf{T}_{l \to l-1})$ is equivalent to

$$\mathbf{T}_L \psi_n^l = 0 \text{ and } \mathbf{T}_R \psi_n^l = 0, \tag{63}$$

for all $n$, and

$$\mathbf{P}_{\mathbf{T}_{l \to l-1}} \psi^l = \{\mathbf{P}_{\mathbf{T}_L} \mathbf{P}_{\mathbf{T}_R} \psi_n^l\}_{\text{all } n} \tag{64}$$

follows.

To define the remaining operators we decompose the Hamiltonian into an onsite and nearest-neighbor terms as

$$h_n = k_{n,n+1} + \frac{1}{2}(v_n + v_{n+1}). \tag{65}$$

In terms of these terms we introduce $\mathbf{L}_n^l$, the Liouvillian restricted to $\mathcal{C}_n^l$,

$$\mathbf{L}_n^l \zeta_n^l = i \sum_{m=n-l/2}^{n+l/2-1} [k_{m,m+1}, \zeta_n^l] + i \sum_{m=n-l/2}^{n+l/2} [v_m, \zeta_n^l]. \tag{66}$$

To further simplify the notation let the super and subscripts on $\mathbf{L}$ be implicit and inferred from the element acted on, e.g.,

$$\mathbf{L}\zeta_n^l = \mathbf{L}_n^l \zeta_n^l. \tag{67}$$

We further introduce the operators $\mathbf{L}_{n,L}^l$ and $\mathbf{L}_{n,R}^l$ for the Liouvillian induced by the nearest-neigbhor terms at the boundaries of $\mathcal{C}_n^l$, defined as

$$\mathbf{L}_{n,L}^l \zeta_n^l = i[k_{n-l/2,n-l/2+1}, \zeta_n^l], \tag{68}$$

$$\mathbf{L}_{n,R}^l \zeta_n^l = i[k_{n+l/2-1,n+l/2}, \zeta_n^l]. \tag{69}$$

Also for these operators we drop the super and subscripts when they can be determined from context. We further introduce the short-hand notation

$$\mathbf{TL}_L = \mathbf{T}_L \mathbf{L}_L. \tag{70}$$

We will make use of the pseudo-inverses $\mathbf{TL}_L^+ \equiv (\mathbf{TL}_L)^+$ and $\mathbf{TL}_R^+ \equiv (\mathbf{TL}_R)^+$. For the specific case of the mixed-field Ising Hamiltonian

$$k_{n,n+1} = J s_n^z s_{n+1}^z, \tag{71}$$

$$v_n = h_L s_n^z + h_T s_n^x, \tag{72}$$

it is possible to derive the following analytical expressions[5]

$$\mathbf{TL}_L^+ = \frac{1}{8J^2}\mathbf{TL}_L^T, \quad \mathbf{TL}_R^+ = \frac{1}{8J^2}\mathbf{TL}_R^T, \tag{73}$$

where $\mathbf{TL}_{L/R}^T \equiv (\mathbf{TL}_{L/R})^T$.

Using these definitions the linear map $\boldsymbol{\Phi}$ in Eq. (37) that gives the derivative $\dot{\Omega}^l$ from $\Omega^{l+1}$ takes a simple form: if $\Psi^l$ is defined as $\Psi^l = \boldsymbol{\Phi}\psi^{l+1}$ then the elements of $\Psi^l$ are

$$\Psi_n^l = \mathbf{L}\psi_n^l + \mathbf{TL}_L\psi_{n-1/2}^{l+1} + \mathbf{TL}_R\psi_{n+1/2}^{l+1}. \tag{74}$$

Recall the convention (54), i.e., by definition $\psi^l = \mathbf{T}_{l+1\to l}\psi^{l+1}$.

We now write $\boldsymbol{\Phi}$ as

$$\boldsymbol{\Phi} = \boldsymbol{\Phi}Q_{\mathbf{T}_{l+1\to l}} + \boldsymbol{\Phi}P_{\mathbf{T}_{l+1\to l}}. \tag{75}$$

The result when the first term $\boldsymbol{\Phi}Q_{\mathbf{T}_{l+1\to l}}$ acts on $\Omega^{l+1}$ can be calculated using only $\Omega^l$, so the intepretation of $\boldsymbol{\Phi}Q_{\mathbf{T}_{l+1\to l}}$ is that it gives the part of the derivative of the $l$-local density matrices which can be deduced from the $l$-local density matrices themselves. The other part, $\boldsymbol{\Phi}P_{\mathbf{T}_{l+1\to l}}$, then gives the unknown part of the derivative of $\Omega^l$. Using the above expressions (74) and (64) we get a simple expression for it: if we define $\Gamma^l$ as $\Gamma^l = \boldsymbol{\Phi}P_{\mathbf{T}_{l+1\to l}}\gamma^{l+1}$, its elements are

$$\Gamma_n^l = \mathbf{TL}_L\mathbf{P}_{\mathbf{T}_R}\gamma_{n-1/2}^{l+1} + \mathbf{TL}_R\mathbf{P}_{\mathbf{T}_L}\gamma_{n+1/2}^{l+1}. \tag{76}$$

Here we used the fact that $\mathbf{TL}_L = \mathbf{TL}_L\mathbf{P}_{\mathbf{T}_L}$ and similar for the operator with subscript $R$.

We now want to write the projector onto the space of what the unknown part of the derivative could be. That is to say we want to write the projector onto the image $\mathrm{im}(\boldsymbol{\Phi}P_{\mathbf{T}_{l+1\to l}})$ of $\boldsymbol{\Phi}P_{\mathbf{T}_{l+1\to l}}$. If $\Gamma^l \in \mathrm{im}(\boldsymbol{\Phi}P_{\mathbf{T}_{l+1\to l}})$ then there are constraints imposed on each of the elements $\{\Psi_n^l\}$ in $\Psi^l$ separately. By an extended derivation it can be shown that the orthogonal projector onto the space fullfilling these constraints is

$$\mathbf{I}_{\boldsymbol{\Phi}P_{\mathbf{T}_{l+1\to l}}}^D = \mathbf{I}_{\mathbf{TL}_L}\mathbf{P}_{\mathbf{T}_R} + \mathbf{I}_{\mathbf{TL}_R}\mathbf{P}_{\mathbf{T}_L} - \mathbf{I}_{\mathbf{TL}_R}\mathbf{I}_{\mathbf{TL}_L}. \tag{77}$$

The superscript $D$ marks that this projector projects onto the "diagonal" constraints imposed by $\Gamma^l \in \mathrm{im}(\boldsymbol{\Phi}P_{\mathbf{T}_{l+1\to l}})$, i.e., the constraints imposed on each of the elements in $\Psi^l$ separately. However, there are also non-diagonal constraints, i.e., if $\Gamma^l \in \mathrm{im}(\boldsymbol{\Phi}P_{\mathbf{T}_{l\to l-1}})$ then the elements $\Gamma_n^l$ and $\Gamma_{n'}^l$ are generally not independent. So, we write the operator $\mathbf{I}_{\boldsymbol{\Phi}P_{\mathbf{T}_{l+1\to l}}}$ as

$$\mathbf{I}_{\boldsymbol{\Phi}P_{\mathbf{T}_{l+1\to l}}} = \mathbf{I}_{\boldsymbol{\Phi}P_{\mathbf{T}_{l+1\to l}}}^{ND}\mathbf{I}_{\boldsymbol{\Phi}P_{\mathbf{T}_{l+1\to l}}}^D, \tag{78}$$

where the operator $\mathbf{I}_{\boldsymbol{\Phi}P_{\mathbf{T}_{l+1\to l}}}^D$ is extended from an operator acting on Hermitian matrices to act on sets of Hermitian matrices, as

$$\mathbf{I}_{\boldsymbol{\Phi}P_{\mathbf{T}_{l\to l-1}}}^D \Gamma^l = \{\mathbf{I}_{\boldsymbol{\Phi}P_{\mathbf{T}_{l\to l-1}}}^D \Gamma_n^l\}_{\text{all } n}. \tag{79}$$

By an extended derivation it can be shown that the operator $\mathbf{I}_{\boldsymbol{\Phi}P_{\mathbf{T}_{l\to l-1}}}^{ND}$, which acts according to the below equation, produces the projector $\mathbf{I}_{\boldsymbol{\Phi}P_{\mathbf{T}_{l+1\to l}}}$ together with $\mathbf{I}_{\boldsymbol{\Phi}P_{\mathbf{T}_{l+1\to l}}}^D$; if $\Sigma^l$ is defined as $\Sigma^l = \mathbf{I}_{\boldsymbol{\Phi}P_{\mathbf{T}_{l\to l-1}}}^{ND}\sigma^l$, then its elements are given by

$$\Sigma_n^l = \frac{1}{2}\mathbf{TL}_R^+\mathbf{TL}_L\sigma_{n-1}^l + \left(\mathbb{1} - \frac{1}{2}(\mathbf{Q}_{\mathbf{TL}_L} + \mathbf{Q}_{\mathbf{TL}_R})\right)\sigma_n^l + \frac{1}{2}\mathbf{TL}_L^+\mathbf{TL}_R\sigma_{n+1}^l. \tag{80}$$

---

[5]For a general nearest neighbor Hamiltonian one has to numerically find the pseudoinverse. Since this operator acts as the identity operator on all but two sites this amounts to finding the pseudoinverse of a $d^2 \times d^4$ matrix ($d$ is the local Hilbert space dimension).



Figure 9: $\mathcal{I}_{\text{tot}}^l$ corresponds to summing over an isosceles trapezoid in the information lattice. As is visualized in the figure, this sum can be recast into a sum over triangles which sum up to the total information corresponding to the neighborhood at the tip of the triangle (12). So we get $\mathcal{I}_{\text{tot}}^l = \sum_n I(\Omega_n^l) - \sum_n' I(\Omega_n^{l-1})$, where $\sum_n'$ indicates that the sum runs over all $n$ except the ones corresponding to the left and the right most neighborhoods.

## B.2 The information-flow derivative

We are now ready to write a closed form expression for the derivative $\dot{\Omega}^l$ in the information flow algorithm. Specifying the derivative $\dot{\Omega}^l$ is equivalent to choosing an element $\chi^l \in \Phi(\mathcal{C}_{\Omega^l}^{l+1})$, where $\mathcal{C}_{\Omega^l}^{l+1}$ is the space of $(l+1)$-local density matrices compatible with $\Omega^l$, see (49). A general element $\psi^{l+1} \in \mathcal{C}_{\Omega^l}^{l+1}$ can be taken to be of the form

$$\psi^{l+1} = \bar{\psi}^{l+1} + \tilde{\psi}^{l+1}, \tag{81}$$

where $\bar{\psi}^{l+1}$ is the minimum norm solution to $\mathbf{T}_{l+1\to l}\bar{\psi}^{l+1} = \Omega^l$ and $\tilde{\psi}^{l+1} \in \ker(\mathbf{T}_{l+1\to l})$. The elements of the minimum norm solution are

$$\bar{\psi}_n^{l+1} = \mathbf{T}_R^+ \Omega_{n-1/2}^l + \mathbf{T}_L^+ \Omega_{n+1/2}^l. \tag{82}$$

We now define $\bar{\chi}^l = \Phi(\bar{\psi}^{l+1})$, and a general $\chi^l \in \Phi(\mathcal{C}_{\Omega^l}^{l+1})$ is thus of the form

$$\chi^l = \bar{\chi}^l + \tilde{\chi}^l, \qquad \tilde{\chi}^l \in \Phi[\ker(\mathbf{T}_{l+1\to l})], \tag{83}$$

with

$$\bar{\chi}_n^l = \mathbf{L}\Omega_n^l + \mathbf{T}_R^+ \mathbf{TL}_L \Omega_{n-1}^l + \mathbf{T}_L^+ \mathbf{TL}_R \Omega_{n+1}^l. \tag{84}$$

Operators with an $R$ subscript commute with operators with an $L$ subscript so their ordering is not important. When operators commute we will use the convention of keeping pseudoinverses furthest to the left.

The idea is now to constrain $\tilde{\chi}^l$ in steps to finally make $\chi^l$ unique. First we constrain $\tilde{\chi}^l$ such that the current condition (47),

$$\mathcal{J}_{l\to l+1} = \frac{\mathcal{I}_l}{\mathcal{I}_{l-1}} \mathcal{J}_{l-1\to l}, \tag{85}$$

is fulfilled. The current $\mathcal{J}_{l\to l+1}$ is

$$\mathcal{J}_{l\to l+1} = -\frac{d}{dt} I_{\text{tot}}^l = -\frac{d}{dt} \sum_{l'=0}^{l} \mathcal{I}^{l'} = \frac{d}{dt}\left(\sum_n S(\Omega_n^l) - \sum_n' S(\Omega_n^{l-1})\right), \tag{86}$$

where the sum $\sum_n'$ indicates that the sum runs over all $n$ except the ones corresponding to the left and the right most neighborhoods. The equality on the second line is explained in Fig. 9. We now write the time-derivatives in terms of the gradient

$$\frac{d}{dt}S(\Omega_n^l) = \langle \dot{\Omega}_n^l | \nabla S(\Omega_n^l) \rangle \,, \tag{87}$$

which has a closed form expression. The function $S(\Omega_{n'}^l)$ can be interpreted both as a function on the space of Hermitian matrices on $\mathcal{C}_{n'}^l$ and as a function on the space of sets of Hermitian matrices. In the first case the gradient is

$$\nabla S(\Omega_{n'}^l) = -\log_2(\Omega_{n'}^l) - \mathbb{1}/\ln(2)\,, \tag{88}$$

and in the second case it is

$$\nabla S(\Omega_{n'}^l) = \{\delta_{n,n'}[-\log_2(\Omega_n^l) - 1/\ln(2)]\}_{\text{all } n}\,. \tag{89}$$

We let it be understood from the context which definition we are using. We then get

$$\mathcal{J}_{l\to l+1} = \langle \Phi(\Omega^l)|\{\log_2(\Omega_n^{l-1})\}_{\text{all}'\, n}\rangle - \langle \tilde{\chi}^l + \bar{\chi}^l|\{\log_2(\Omega_n^l)\}_{\text{all } n}\rangle\,. \tag{90}$$

Here "all'" has an analogous meaning as $\sum_n'$ in (86): it means all $n$ except the ones corresponding to the left and the right most neighborhoods (those elements of the set are instead taken to be zero).

From this rewriting of the current (and the analogous rewriting for $\mathcal{J}_{l-1\to l}$) it follows that complying with the current-condition (47) amounts to setting the inner-product $\langle \tilde{\chi}^l|\{\log_2(\Omega_n^l)\}_{\text{all } n}\rangle$ equal to a $\Omega^l$ dependent constant,

$$\langle \tilde{\chi}^l|\{\log_2(\Omega_n^l)\}_{\text{all } n}\rangle = \alpha(\Omega^l)\,, \tag{91}$$

which takes the form

$$\alpha(\Omega^l) = \langle \Phi(\Omega^l)|\{\log_2(\Omega_n^{l-1})\}_{\text{all}'\, n}\rangle - \langle \bar{\chi}^l|\{\log_2(\Omega_n^l)\}_{\text{all } n}\rangle$$
$$+ \mathcal{I}_l\mathcal{I}_{l-1}^{-1}\Big(\langle \Phi(\Omega^l)|\{\log_2(\Omega_n^{l-1})\}_{\text{all } n}\rangle - \langle \Phi(\Omega^{l-1})|\{\log_2(\Omega_n^{l-2})\}_{\text{all}'\, n}\rangle\Big)\,. \tag{92}$$

So we can now write the expression for a general $\tilde{\chi}^l$ with the current condition fulfilled,

$$\tilde{\chi}^l = \bar{\bar{\chi}}^l + \tilde{\tilde{\chi}}^l\,, \qquad \tilde{\tilde{\chi}}^l \in S_\perp\,, \tag{93}$$

where

$$S_\perp = \{\chi \in \Phi(\ker(\mathbf{T}_{l+1\to l}))|\langle \chi|\{\log_2(\Omega_n^l)\}_{\text{all } n}\rangle = 0\}\,, \tag{94}$$

and

$$\bar{\bar{\chi}}^l = \bar{\chi}^l + \frac{\alpha(\Omega^l)\mathbf{I}_{\Phi\mathbf{P}_{\mathbf{T}_{l\to l-1}}}\{\log_2(\Omega_n^l)\}_{\text{all } n}}{\left\langle \{\log_2(\Omega_n^l)\}_{\text{all } n} \middle| \mathbf{I}_{\Phi\mathbf{P}_{\mathbf{T}_{l\to l-1}}}\{\log_2(\Omega_n^l)\}_{\text{all } n} \right\rangle}\,. \tag{95}$$

However, to specify $\chi^l$ fully we need to constrain $\tilde{\chi}^l$ further. We use the prescription from the main text and choose $\tilde{\tilde{\chi}}^l$ (the degrees of freedom which do not affect the current condition) by minimizing $b_{\Omega^l}(\chi, \chi)$ in (50), i.e.,

$$I_{\text{tot}}^l(\Omega^l + \epsilon\chi) = I_{\text{tot}}^l(\Omega^l) - \epsilon\mathcal{J}_{l\to l+1}(\chi) + \frac{\epsilon^2}{2}b_{\Omega^l}(\chi, \chi) + \mathcal{O}(\epsilon^3)\,. \tag{96}$$

We can write $b_{\Omega^l}(\chi,\chi)$ as

$$b_{\Omega^l}(\chi,\chi) \propto \langle \tilde{\tilde{\chi}} | \mathbf{H}_{I_{tot}^l} | \tilde{\tilde{\chi}} \rangle + 2 \langle \tilde{\tilde{\chi}} | \mathbf{H}_{I_{tot}^l} | \bar{\bar{\chi}} \rangle + \text{const.}, \tag{97}$$

where $\mathbf{H}_{I_{tot}^l}$ is the Hessian of $I_{tot}^l$, as a function of $\Omega^l$, and "const." denote terms independent of $\tilde{\tilde{\chi}}$. If there is a unique solution $\tilde{\tilde{\chi}}$, to the equation

$$\mathbf{P}_{S_\perp} \mathbf{H}_{I_{tot}^l} \mathbf{P}_{S_\perp} \tilde{\tilde{\chi}} = \mathbf{P}_{S_\perp} \mathbf{H}_{I_{tot}^l} \bar{\bar{\chi}}^l, \tag{98}$$

then this solution will be the unique minimizer of $b_{\Omega^l}$. The projector $\mathbf{P}_{S_\perp}$ acts in a way which is easy to implement numerically: when acting on any set of matrices $\zeta^l$ it acts as

$$\mathbf{P}_{S_\perp} \zeta^l = \mathbf{I}_{\Phi \mathbf{P}_{\mathbf{T}_{l\to l-1}}} \zeta^l - \mathbf{I}_{\Phi \mathbf{P}_{\mathbf{T}_{l\to l-1}}} \{\log_2(\Omega_n^l)\}_{\text{all } n} \times \frac{\langle \{\log_2(\Omega_n^l)\}_{\text{all } n} | \mathbf{I}_{\Phi \mathbf{P}_{\mathbf{T}_{l\to l-1}}} \zeta^l \rangle}{\langle \{\log_2(\Omega_n^l)\}_{\text{all } n} | \mathbf{I}_{\Phi \mathbf{P}_{\mathbf{T}_{l\to l-1}}} \{\log_2(\Omega_n^l)\}_{\text{all } n} \rangle}. \tag{99}$$

We now discuss how to solve such a linear equation numerically. If one can construct a good conditioning matrix a linear system

$$AX = B, \tag{100}$$

can be solved using the preconditioned conjugate gradient method, see e.g., Ref. [29]. One can then get a solution of the linear equation with numerical resources of the same order of magnitude as it takes to apply the operator $A$ to an element. A conditioning matrix $M$ is a good approximation to the inverse $M \approx A^{-1}$ which can be applied using the same numerical resources as applying $A$ itself. We here use the pedestrian definition of "good" to simply mean that the preconditioned conjugate gradient method converges in only a few ($\lesssim 10$) steps. Using the equation

$$I_{tot}^l(\Omega^l) = \sum_n' S(\Omega_n^{l-1}) - \sum_n S(\Omega_n^l), \tag{101}$$

we see that the Hessian $\mathbf{H}_{I_{tot}^l}$ is

$$\mathbf{H}_{I_{tot}^l} = \mathbf{H}_{\sum_n' S(\Omega_n^{l-1})} - \mathbf{H}_{\sum_n S(\Omega_n^l)}. \tag{102}$$

However $\Phi(\ker(\mathbf{T}_{l+1\to l})) \subset \ker(\mathbf{T}_{l\to l-1})$, so elements in $\Phi(\ker(\mathbf{T}_{l+1\to l}))$ do not alter the $(l-1)$-local density matrices, and we get

$$\mathbf{P}_{S_\perp} \mathbf{H}_{I_{tot}^l} \mathbf{P}_{S_\perp} = -\mathbf{P}_{S_\perp} \mathbf{H}_{\sum_n S(\Omega_n^l)} \mathbf{P}_{S_\perp}. \tag{103}$$

The Hessian of the sum of entropies $\sum_n S(\Omega_n^l)$ can be expanded as a sum of Hessians of the entropy of each density matrix $\Omega_n^l$,

$$\mathbf{H}_{\sum_n S(\Omega_n^l)} = \sum_n \mathbf{H}_{S(\Omega_n^l)}. \tag{104}$$

Analogous to the situations with the gradients, the Hessians are either functions of Hermitian matrices or of sets of Hermitian matrices, depending on if the function $S(\Omega_{n'}^l)$ is interpreted as a function on the space of Hermitian matrices on $\mathcal{C}_{n'}^l$ or as a function on the space of sets of Hermitian matrices. This means that

$$\mathbf{H}_{S(\Omega_{n'}^l)} \zeta^l = \{\delta_{n,n'} \mathbf{H}_{S(\Omega_{n'}^l)} \zeta_n^l\}_{\text{all } n}, \tag{105}$$

where $\mathbf{H}_{S(\Omega_{n'}^l)}$ on the left hand side is the Hessian when $S(\Omega_{n'}^l)$ is interpreted as a function on the space of sets of Hermitian matrices and $\mathbf{H}_{S(\Omega_{n'}^l)}$ on the right hand is the Hessian when

$S(\Omega^l_{n'})$ is interpreted as a function of Hermitian matrices. As with the gradients, which one we are referring to can be understood from the context.

The entropy can be written purely in terms of the eigenvalues $\{\kappa^l_{n,i}\}_{i=1,\dots,\dim(\Omega^l_n)}$ of $\Omega^l_n$,

$$S(\Omega^l_n) = -\sum_i \kappa^l_{n,i} \log_2(\kappa^l_{n,i}), \tag{106}$$

so the Hessian can be written in terms of the well-known formulas for the series expansion of the eigenvalues (i.e., the perturbation theory formulas). The result, when $\mathbf{H}_{S(\Omega^l_n)}$ acts on any zero-trace matrix $\zeta^l_n$ is

$$\mathbf{H}_{S(\Omega^l_n)}\zeta^l_n = U^l_n(\mathcal{H}^D_{S(\Omega^l_n)} * U^{l\dagger}_n \zeta^l_n U^l_n)U^{l\dagger}_n, \tag{107}$$

where $*$ denotes elementwise multiplication, and $U^l_n$ is the matrix which has the eigenvectors of $\Omega^l_n$ as rows and $\mathcal{H}^D_{S(\Omega^l_n)}$ is the matrix with elements

$$[\mathcal{H}^D_{S(\Omega^l_n)}]_{i,j} = -\frac{1}{\ln(2)} \frac{\operatorname{arctanh}\left(\frac{\kappa^l_{ni}-\kappa^l_{nj}}{\kappa^l_{ni}+\kappa^l_{nj}}\right)}{\kappa^l_{ni}-\kappa^l_{nj}}. \tag{108}$$

Note that since $\operatorname{arctanh}(x) = x + \mathcal{O}(x^2)$ the above expression is well-defined also for the diagonal elements $[\mathcal{H}^D_{S(\Omega^l_n)}]_{i,i}$ or degeneracies of the eigenvalues $\{\kappa^l_{ni}\}$. By direct inspection, we see that the eigenvalues of the operator $\mathbf{H}_{S(\Omega^l_n)}$ are $\{[\mathcal{H}^D_{S(\Omega^l_n)}]_{i,j}\}_{\text{all } i,j}$, which are all strictly negative if all eigenvalues $\{\kappa^l_{n,i}\}$ are strictly positive. So if we assume that all density matrices $\{\Omega^l_n\}_{\text{all } n}$ are positive definite then it follows from (104) that $\mathbf{H}_{\sum_n S(\Omega^l_n)}$ is negative definite. In turn, this means that

$$\mathbf{P}_{S_\perp}\mathbf{H}_{I^l_{tot}}\mathbf{P}_{S_\perp} = -\mathbf{P}_{S_\perp}\mathbf{H}_{\sum_n S(\Omega^l_n)}\mathbf{P}_{S_\perp}, \tag{109}$$

restricted to $S_\perp$ is positive definite which means that there is a unique solution to the equation (98) which defines $\tilde{\tilde{\chi}}$.

From the above expression (107) for the Hessian of the entropy we can also write an analytical expression for how the inverse $\mathbf{H}^{-1}_{S(\Omega^l_n)}$ acts:

$$\mathbf{H}^{-1}_{S(\Omega^l_n)}\zeta^l_n = U^l_n(\mathcal{H}^D_{S(\Omega^l_n)}{}^{*-1} * U^{l\dagger}_n \zeta^l_n U^l_n)U^{l\dagger}_n, \tag{110}$$

where $\mathcal{H}^D_{S(\Omega^l_n)}{}^{*-1}$ denotes elementwise inversion of $\mathcal{H}^D_{S(\Omega^l_n)}$.

In general $\mathbf{H}_{\sum_n S(\Omega^l_n)}$ and $\mathbf{P}_{S_\perp}$ does not commute, so $\mathbf{M}\bar{\bar{\chi}}^l$ with

$$\mathbf{M} = \mathbf{P}_{S_\perp}\mathbf{H}^{-1}_{\sum_n S(\Omega^l_n)}\mathbf{P}_{S_\perp}, \tag{111}$$

is not a solution to linear equation (98) which defines $\tilde{\tilde{\chi}}$. However, at least in the examples we have considered in this paper, $\mathbf{M}$ makes a good conditioning matrix, allowing us to efficiently find the solution numerically.

## C  The Petz recovery map algorithm

We have already discussed the basics of the Petz recovery map algorithm: if all $i^{l+1}_n$ are sufficiently small then one can use the Petz recovery map to calculate the $(l+1)$-local density

matrices given the $l$-local density matrices, making the time-evolution closed. The purpose of this section is to precisely define how we do this.

If the conditional mutual information vanishes, $I(A;B|C) = 0$, there are several Petz recovery maps, i.e., several analytical expressions for expressing a density matrix on three parts $\rho_{ABC}$ in terms of the corresponding reduced density matrices $\rho_{AB}$ and $\rho_{BC}$. In fact, if $I(A;B|C) = 0$ the three below expressions all equal to $\rho_{ABC}$,

$$\rho_{ABC} = \rho_{AB}^{1/2} \rho_B^{-1/2} \rho_{BC} \rho_B^{-1/2} \rho_{AB}^{1/2} \tag{112}$$

$$= \rho_{BC}^{1/2} \rho_B^{-1/2} \rho_{AB} \rho_B^{-1/2} \rho_{BC}^{1/2} \tag{113}$$

$$= \exp\left(\ln \rho_{AB} + \ln \rho_{BC} - \ln \rho_B\right). \tag{114}$$

As we have mentioned, only the last of these maps (114) has a well-known bound on the error, when $I(A;C|B) \neq 0$. In practice, we have found that the other two maps are nonetheless better, and their numerical implementations are faster. As a first approximation of $\rho_{ABC}$ we use

$$\tilde{\varrho}_{ABC} = \begin{cases} \rho_{AB}^{1/2} \rho_B^{-1/2} \rho_{BC} \rho_B^{-1/2} \rho_{AB}^{1/2}, & \text{if } I(B;C) > I(A;B), \\ \rho_{BC}^{1/2} \rho_B^{-1/2} \rho_{AB} \rho_B^{-1/2} \rho_{BC}^{1/2}, & \text{if } I(B;C) < I(A;B), \end{cases} \tag{115}$$

and if $I(B;C) = I(A;B)$ we average over the above two choices. If $I(A;C|B) \neq 0$ then this approximation does not necessarily preserve $\rho_{AB}$ and $\rho_{BC}$, so we add a projection step and write the final approximation, $\varrho_{ABC}$, of $\rho_{ABC}$ as

$$\varrho_{ABC} = \tilde{\varrho}_{ABC} + (\rho_{AB} - \tilde{\varrho}_{AB}) \otimes I_2 + I_2 \otimes (\rho_{BC} - \tilde{\varrho}_{BC}) - I_2 \otimes (\rho_B - \tilde{\rho}_B) \otimes I_2, \tag{116}$$

where

$$\tilde{\varrho}_{BC} = \operatorname*{Tr}_A \tilde{\varrho}_{ABC}, \quad \tilde{\varrho}_{AB} = \operatorname*{Tr}_C \tilde{\varrho}_{ABC}. \tag{117}$$

This expression is the orthogonal projection of $\tilde{\varrho}_{ABC}$ onto the space of density matrices which have $\rho_{AB}$ and $\rho_{BC}$ as partial traces.

The approximation $\varrho_{ABC}$ of $\rho_{ABC}$ provides an approximation of the $(l+1)$-local density matrices, given the $l$-local density matrices. For example, if we take $AB = \mathcal{C}_{n-1/2}^l$ and $BC = \mathcal{C}_{n+1/2}^l$ then $\varrho_{ABC}$ approximates $\rho_{\mathcal{C}_n^{l+1}}$ given $\rho_{\mathcal{C}_{n-1/2}^l}$ and $\rho_{\mathcal{C}_{n+1/2}^l}$.

# D  Integration schemes

## D.1  Runge-Kutta methods

In this work we integrate all differential equations with Runge-Kutta methods, that is,

$$\Omega^l(t + \Delta t) = \Omega^l(t) + \Delta t \sum_{i=1}^{K} b_i \kappa^{l,i} + \mathcal{O}(\Delta t^N), \qquad \kappa^{l,i} = \Psi\left(\Omega^l(t) + \Delta t \sum_{j=1}^{i-1} a_{ij} \kappa^{l,j}\right), \tag{118}$$

where $\Psi$ is one of the compatible derivative functions (40) and $\{b_i\}$ and $\{a_{ij}\}$ are Runge-Kutta parameters. We use the parameters[6] from Ref. [30] with a step-size error of $\mathcal{O}(\Delta t^{12})$. We also use a dynamic step-size [31] ensuring a step-size error smaller than $10^{-5}$.

---

[6]Files with the parameters of this Runge-Kutta method as well as other high order methods of the same type can be found at sce.uhcl.edu/rungekutta/.

In a numerically more demanding situations one would want to allow for a bigger step-size error to allow for faster runtimes. It is worth noting that this does not affect conservation of constants of the motion. Since $\mathbf{\Psi}$ is compatible it follows that the expectation values

$$\left\langle \kappa^{l,i} \,\middle|\, \omega^l \right\rangle = 0 \,, \tag{119}$$

of any constant of motion $\mathcal{O}$ of the form

$$\mathcal{O} = \sum_n \omega_n^l \,, \tag{120}$$

is zero for all $\kappa^{l,i}$. It follows that expectation value of all constants of motion are exactly the same for $\Omega^l(t + \Delta t)$ and $\Omega^l(t)$ (no matter the value of $\Delta t$).

## D.2 Dealing with small eigenvalues

If some of the matrices in the set $\Omega^l(t)$ have small eigenvalues, then one of the intermediate values

$$\Omega^l(t) + \Delta t \sum_{j=1}^{i-1} a_{ij} \kappa^{l,j} \,, \tag{121}$$

could have matrices with negative eigenvalues. The functions $\mathbf{\Psi}$ we consider are defined only for semi-positive definite matrices, and the Runge-Kutta methods can therefore fail in this case. In the simulations in this paper this is not a problem. There are no small eigenvalues in the case with the translational invariant initial state (35). For the initial state (36) there are initially matrices with vanishing eigenvalues, but these can be dealt with as follows. We first shift the state $\rho(t)$ with the maximally mixed state to form $\rho_{\text{shift}}(t)$. Since the full Schrödinger equation is linear, we can time-evolve this shifted state and at a later time $t'$ shift back,

$$\rho_{\text{shift}}(t) = \frac{1}{2} \left[ \rho(t) + \dim(\rho)^{-1}\mathbb{1} \right] \Longleftrightarrow \tag{122}$$

$$\rho(t') = 2\rho_{\text{shift}}(t') - \dim(\rho)^{-1}\mathbb{1} \,. \tag{123}$$

For the local density matrices this shift amounts to

$$\Omega_{\text{shift}}^l = \{\frac{1}{2}[\Omega_n^l + \dim(\Omega_n^l)^{-1}\mathbb{1}]\}_{\text{all } n} \,, \tag{124}$$

where $\Omega^l = \{\Omega_n^l\}_{\text{all } n}$ is the unshifted $l$-local density matrices. If the function $\mathbf{\Psi}$ which estimates the derivative gives an equally good estimate (i.e., converges equally fast as a function of $l$) for the derivative of $\Omega_{\text{shift}}^l$ as it does for $\Omega^l$, we can just as well time-evolve $\Omega_{\text{shift}}^l$ and then shift back. This is the case when using the Petz algorithm for the simulation with the initial state (36). However, there is in general no guarantee that the estimates $\mathbf{\Psi}$ for the derivatives converge as quickly with $l$ for the shifted case, as for the unshifted, requiring a larger truncation than if the unshifted local density matrices could be time-evolved directly. To solve the general situation of small eigenvalues one must instead use a different integration scheme. The smallest eigenvalues generically increase when there is a flow of information from small to large scales. So it is only either early in the time-evolution or in situations where there is no flow of information to larger scales where such an integration scheme is needed. In both these situations we can use the Petz-recovery map algorithm and then we have access to a function $\mathbf{E}$ of the $l$-local density matrices $\Omega^l$ which approximates the $(l + 1)$-local density matrices,

$$\Omega^{l+1} \approx \mathbf{E}(\Omega^l) \,. \tag{125}$$

If one knows the $(l+1)$-local density matrices of a state $\rho$, one can calculate the $l$-local density matrices of the state

$$e^{iA_{n,n+1}}\rho\,e^{-iA_{n,n+1}}\,, \tag{126}$$

where $A_{n,n+1}$ is any operator acting on sites $n$ and $n+1$. So, the function $\mathbf{E}$ provides a prescription of how to act with any function of the form $e^{iA_{n,n+1}}$ on $\Omega^l$. Using the Suzuki-Trotter decomposition, see e.g., [32], we can write the time-evolution operator

$$e^{i\Delta t H} = \prod_{k=1}^{K}\left(\prod_{n\,\text{odd}} e^{i\Delta t\,\alpha_k h_{n,n+1}}\right)\left(\prod_{n\,\text{even}} e^{i\Delta t\,\beta_k h_{n,n+1}}\right) + \mathcal{O}(\Delta t^N)\,, \tag{127}$$

where $\{\alpha_k, \beta_k\}$ are parameters which can be chosen to make $N$ arbitrarily large at the cost of a larger order $K$. We can then use above prescription for acting with an operator of the form $e^{iA_{n,n+1}}$ to act with every factor in this this expansion, and thus get an approximation for $\Omega^l(t+\Delta t)$ from $\Omega^l(t)$. This integration method has no problems with positivity, and can thus be used also when there are small or vanishing eigenvalues. However, when possible it is advantageous to use Runge-Kutta methods. The first reason is that for the same order of the approximation $N$ the Suzuki-Trotter decomposition typically requires more steps $K$ than the the best Runge-Kutta method for the same $N$. This means that one has to apply $\mathbf{E}$ more times, which is the most numerically demanding part of the algorithm. Furthermore, for the Runge-Kutta integration there is no time-step error in constants of motion, but for the Suzuki Trotter integration, constants of motion are on the same footing as everything else. Typically, errors in constants of motion are more severe than errors in other operators, and therefore one typically requires a smaller time-step error when using Suzuki-Trotter integration.

## D.3   Infinite systems

We address the question of how to integrate the local density matrices in an infinite system. When we have translation symmetry this is straightforward. If $\Omega_n^l = \Omega_{n+k}^l$ and we only have to keep track of the $k$ density matrices $\tilde{\Omega}^l = \{\Omega_n^l\}_{n=1,\dots,k}$. A function $\Psi(\tilde{\Omega}^l)$ which approximates the time-derivative of $\tilde{\Omega}^l$ is straightforwardly inherited from the definition of $\Psi$ for a finite space.

The initial condition (36),

$$\rho_{t=0} = \cdots \otimes I_2 \otimes I_2 \otimes |\uparrow_x\rangle\langle\uparrow_x| \otimes I_2 \otimes I_2 \otimes \cdots\,, \tag{128}$$

is however not translation invariant, requiring some care. As before we use $n_0$ to denote the site where the spin initially pointed up in the $s_x$ direction. At any finite time $t$ there will be some finite length $\Lambda(t)$ such that with high precision

$$\rho_{[n_0+\Lambda,n_0+l+\Lambda]} \approx \rho_{[n_0+\Lambda,n_0+\Lambda+l-1]} \otimes I_2\,, \tag{129}$$

and similarly

$$\rho_{[n_0-\Lambda-l,n_0-\Lambda]} \approx I_2 \otimes \rho_{[n_0-\Lambda-l+1,n_0-\Lambda]}\,, \tag{130}$$

on the left. So up to time $t$ we only need to consider a finite number, $2\Lambda+l-1$, of local density matrices and define the time-derivative by assuming that the rest are given by tensor products as in (129).

To utilize this we start out with $\Omega^l(0)$ consisting of the $2\Lambda_0+l-1$ density matrices centered around $n_0$. Before the first time-step we add $k$ sites on either side using (129). We then time-evolve a finite time step $\Delta t$ and afterwards remove from $\Omega^l(\Delta t)$ all density matrices which can be approximated by (129) with a given error $\epsilon$, i.e., we remove the density matrix $\rho_{[n,n+l]}$ if

$$\text{Tr}\left(\rho_{[n,n+l-1]} \otimes I_2 - \rho_{[n,n+l]}\right)^2 < \epsilon^2\,. \tag{131}$$

If we remove no density matrix we have kept track of too few density matrices for the approximation (129) to be valid, and need to redo the time-step with a larger $k$. If we removed some density matrices we end up with $\Omega^l(\Delta t)$ consisting of $2\Lambda_1 + l - 1$ with $\Lambda_1 \geq \Lambda_0$. We then continue the procedure of first adding density matrices then making a time step and removing density matrices. The number of elements in $\Omega^l(t)$ we keep track of then grows, with accompanying growth of the numerical resources required to do a time-step. For the time-evolution we focussed on in the main text the growth of the number of elements is asymptotically constrained by the energy diffusion and the number of elements (and thus the numerical resources) grows as $\sqrt{t}$.

### D.4 Utilizing discrete symmetries

If the system under consideration has a unitary symmetry, one can in general use it to reduce the numerical resources required to time-evolve the local density matrices. For the simulation with initial state (35) we use reflection symmetry to speed up the time-evolution.

By unitary symmetry we mean that the Hamiltonian commutes with an unitary operator $[U, H] = 0$. If a state $\rho(t)$ satisfies this symmetry at a given time $t$, i.e.,

$$\rho(t) = U\rho(t)U^{-1}, \tag{132}$$

then it will satisfy it for all times. The above equality manifests itself by a corresponding relation for the local density matrices

$$\Omega^l = f_U(\Omega^l). \tag{133}$$

For example, if $U$ is translation by one site, then (132) implies

$$\Omega^l_n = \Omega^l_{n'}, \qquad \forall n, n'. \tag{134}$$

The opposite is not necessarily true, if $\Omega^l$ satisfies the constraint (133), it does not necessarily imply that the full state upholds the corresponding symmetry (132). Even if all density matrices of scale $l$ are equal the state could still differ on scale $l + 1$. Discrete symmetries are therefore not automatically built into the compatibility condition of the time-derivative (41). So, if there is a symmetry, we can use it to reduce the numerical resources required. Translation invariance is straightforward to utilize. In particular, translation invariance by one site means that all density matrices are equal and we do not have to keep track of a set of density matrices, we only need to keep track of one.

Apart from translation symmetry the only other symmetry we utilize in this paper is reflection symmetry. In the simulation with the translational invariant initial state (35) we have reflection symmetry around every point. This means that every density matrix for all $l$ and $n$ satisfies

$$\Omega^l_n = R\Omega^l_n R^\dagger, \tag{135}$$

where $R$ is the operator which changes the direction of the spatial axes, e.g., on product states in $\mathcal{C}^l_n$ it acts as

$$R|x_{n-l/2}\rangle \otimes |x_{n-l/2+1}\rangle \otimes \cdots |x_{n+l/2}\rangle = |x_{n+l/2}\rangle \otimes \cdots |x_{n-l/2+1}\rangle \otimes |x_{n-l/2}\rangle . \tag{136}$$

This means that

$$\Omega^l_n = \Omega^{l,+}_n + \Omega^{l,-}_n, \tag{137}$$

where $\Omega^{l,+}_n$ ($\Omega^{l,-}_n$) is an operator in the space of states with $R$-eigenvalue 1 ($-1$). Knowing this form of the density matrix allows for roughly four times faster diagonalization of $\Omega^l_n$ and subsequently a faster evaluation of $\Psi$.

# E  $l$-local Gibbs states

An $l$-local Gibbs state, $\rho^l_{\text{Gibbs}}$, is the maximum entropy state with given $l$-local density matrices $\Omega^l_{\text{Gibbs}}$. An example is a usual Gibbs state, which is a maximum entropy state given a set of expectation values of local constants of the motion. Also the generalization of the usual Gibbs states to have spatially dependent generalized forces are $l$-local Gibbs states; e.g., a state with spatially varying temperature,

$$\rho(\{\beta_n\}) = \frac{e^{-\sum_n \beta_n h_n}}{\text{Tr}\left(e^{-\sum_n \beta_n h_n}\right)} \,. \tag{138}$$

To see that this complies with the definition of an $l$-local Gibbs state we can imagine making a small change to this state, to form the density matrix $\rho(\{\beta_n\}) + \mathcal{E}$. The entropy then changes as

$$S(\rho(\{\beta_n\}) + \mathcal{E}) = S(\rho(\{\beta_n\})) - \sum_n \beta_n \, \text{Tr}(\mathcal{E} h_n) + \mathcal{O}(\mathcal{E}^2) \,. \tag{139}$$

Here we assumed $\text{Tr}\,\mathcal{E} = 0$, otherwise $\rho(\{\beta_n\}) + \mathcal{E}$ would not have unit trace. Now if $\rho(\{\beta_n\}) + \mathcal{E}$ should have the same reduced density matrices on every pair of consecutive sites, we must have

$$\underset{[n,n+1]^c}{\text{Tr}} (\mathcal{E}) = 0 \,, \qquad n \in \text{ sites} \,. \tag{140}$$

This means that $\text{Tr}(\mathcal{E} h_n) = 0$ and we can conclude that, to first order in $\mathcal{E}$, $\rho(\{\beta_n\}) + \mathcal{E}$ and $\rho(\{\beta_n\})$ have the same entropy. Since the entropy is convex it follows that $\rho(\{\beta_n\})$ is the maximum entropy state given the $l$-local density matrices. It is straightforward to generalize this argument and show that any density matrix $\rho \propto e^{-\mathcal{O}}$, for some operator

$$\mathcal{O} = \sum_n \omega^l_n \,, \quad \omega^l_n \text{ acts on } \mathcal{C}^l_n \,, \tag{141}$$

is an $l$-local Gibbs state.

This argument can also be used in reverse to show that any $l$-local Gibbs state can be cast in the form $\rho \propto e^{-\mathcal{O}}$, for some operator $\mathcal{O}$ as above. If $\rho^l_{\text{Gibbs}}$ is an $l$-local Gibbs state, then the inner-product of the gradient of the entropy with any perturbation $\mathcal{E}$ of $\rho^l_{\text{Gibbs}}$, not changing $l$-local density matrices, must be zero. That is,

$$\text{Tr}(\mathcal{E} \ln(\rho^l_{\text{Gibbs}})) = 0 \,, \tag{142}$$

for all Hermitian matrices with

$$\mathcal{E} \in \ker(\mathbf{T}_{\to l}) \,, \tag{143}$$

where $\mathbf{T}_{\to l}$ is the trace operator which takes a density matrix on the full space and maps it to the corresponding $l$-local density matrix. Equivalently

$$\underset{(\mathcal{C}^l_n)^c}{\text{Tr}} (\mathcal{E}) = 0 \,, \qquad n \in \text{sites} \,. \tag{144}$$

So, since $\text{Tr}(\mathcal{E} \ln(\rho^l_{\text{Gibbs}})) = 0$, the logarithm $\ln(\rho^l_{\text{Gibbs}})$ is an element in the orthogonal complement to the kernel $\ker(\mathbf{T}_{\to l})$: $\ln(\rho^l_{\text{Gibbs}}) \in \perp \ker(\mathbf{T}_{\to l})$. From the expression (144) of the kernel $\ker(\mathbf{T}_{\to l})$ it follows that $\perp \ker(\mathbf{T}_{\to l})$ is spanned by operators of the kind $\omega^l_n$ where $\omega^l_n$ act as identity outside $\mathcal{C}^l_n$. So,

$$\ln(\rho^l_{\text{Gibbs}}) = \sum_n \omega^l_n \,, \qquad \omega^l_n \text{ acts on } \mathcal{C}^l_n \,, \tag{145}$$

which concludes the proof.

### E.1 An algorithm to calculate the reduced density matrices in an $l$-local Gibbs state

In this section we show how to numerically obtain the $k$-local density matrices in an $l$-local Gibbs state, if one has access to the $l$-local density matrices. By definition an $l$-local Gibbs state is the state which minimize the total information

$$I_{\text{tot}} = \sum_{l=0}^{\infty} \mathcal{I}^l \,, \tag{146}$$

given some local density matrices $\Omega_{\text{Gibbs}}^l$. The idea is now to instead minimize the truncated total information

$$I_{\text{tot}}^\lambda = \sum_{l'=0}^{\lambda} \mathcal{I}^{l'} \,. \tag{147}$$

From Kim's inequality

$$\text{Tr}\sqrt{(\rho_{AB} - \sigma_{AB})^2} \le 2\sqrt{I_\rho(A;B) + I_\sigma(A;B)} \,, \tag{148}$$

one can conclude that the difference between $k$-local density matrices gotten from minimizing $I_{\text{tot}}^\lambda$ and the error in $\Omega_{\text{Gibbs}}^k$ (defined by minimizing $I_{\text{tot}}$) is bounded by $\max_m(i_m^{\lambda+1})$. However one can also estimate the error by comparing the minimization of $I_{\text{tot}}^\lambda$ and $I_{\text{tot}}^{\lambda-1}$ and typically the error is much smaller than that given by Kim's inequality.

As we discussed, an $l$-local Gibbs state is of the form

$$\rho_{\text{Gibbs}}^l = e^{-\sum_n \omega_n^l} \,, \tag{149}$$

for some operators $\omega_n^l$ that only act on sites $\mathcal{C}_n^l$. Unless $\rho_{\text{Gibbs}}^l$ is a critical ground-state of $\mathcal{O} = \sum_n \omega_n^l$, $i_m^L$ decays exponentially as a function of $L$. For the minimization done to get the data in Fig. 5, this fast decay meant that we could let $\lambda$ be large enough for the error to be limited only by machine-size precision.

Then comes the next question, how does one minimize $I_{\text{tot}}^\lambda$. We begin by discussion the case when $\lambda = l+1$. We first need a starting point, $\tilde{\Omega}^{l+1}$, that is some $(l+1)$-local density matrices $\tilde{\Omega}^{l+1}$ with the property that $\mathbf{T}_{l+1\to l}\tilde{\Omega}^{l+1} = \Omega_{\text{Gibbs}}^l$. To get a starting point we use the Petz recovery maps as in App. C to get an approximation $\Omega_{\text{Petz}}^{l+1}$.

The Hessian of $I_{\text{tot}}^\lambda$ can be written in terms of Hessians of sums of entropies (102),

$$\mathbf{H}_{I_{tot}^\lambda} = \mathbf{H}_{\sum'_n S(\Omega_n^{\lambda-1})} - \mathbf{H}_{\sum_n S(\Omega_n^\lambda)} \,. \tag{150}$$

Since we are keeping the $l$-local density matrices fixed, we are only after the Hessian restricted to $\ker(\mathbf{T}_{\lambda\to l})$, and as we explained in Sec. B.2 for $\lambda = l+1$ the first term in (102) vanishes, leaving us with

$$\mathbf{P}_{\mathbf{T}_{\lambda\to l}}\mathbf{H}_{I_{tot}^\lambda}\mathbf{P}_{\mathbf{T}_{\lambda\to l}} = -\mathbf{P}_{\mathbf{T}_{\lambda\to l}}\mathbf{H}_{\sum_n S(\Omega_n^\lambda)}\mathbf{P}_{\mathbf{T}_{\lambda\to l}} \,. \tag{151}$$

Since $-\mathbf{H}_{\sum_n S(\Omega_n^\lambda)}$ is positive definite, it follows that $\mathbf{H}_{I_{tot}^\lambda}$ restricted to $\ker(\mathbf{T}_{\lambda\to l})$ also is positive definite. In Sec. B.2 we also showed how to solve linear equations involving $\mathbf{H}_{\sum_n S(\Omega_n^\lambda)}$. In particular we can solve

$$\mathbf{P}_{\mathbf{T}_{\lambda\to l}}\mathbf{H}_{\sum_n S(\Omega_n^l)}\mathbf{P}_{\mathbf{T}_{\lambda\to l}}\zeta^l = \mathbf{P}_{\mathbf{T}_{\lambda\to l}}\nabla I_{\text{tot}}^\lambda \,, \tag{152}$$

meaning that we can use Newton-Raphson's method to find the minimum of $I_{\text{tot}}^\lambda$.

If $\lambda = l + 2$ then we start by using the algorithm above to find the $\Omega^{l+1}$ which minimize $I_{\text{tot}}^{l+1}$. We then extend this as before, using the Petz recovery maps, to get a starting point $\tilde{\Omega}^{l+2}$, i.e., some $(l + 2)$-local density matrices with the property $\mathbf{T}_{l+2 \to l} \tilde{\Omega}^{l+1} = \Omega_{\text{Gibbs}}^l$.

For $\lambda > l + 1$, the first term in the expression (102) for the Hessian $\mathbf{H}_{I_{tot}^\lambda}$ does not vanish when restricted to $\ker(\mathbf{T}_{\lambda \to l})$. When both terms are present there is no guarantee that the Hessian is positive definite; $I_{\text{tot}}^\lambda$ is in general not convex. However for a maximally mixed set of density matrices it is positive definite and smooth. So we expect that this only is a problem for density matrices with very small eigenvalues. For the minimization done to get the data in Fig. 5, the Hessian has been positive definite close to the starting points $\tilde{\Omega}^{l+2}$ and we have been able to use Newton-Raphson's method to find the minimum closest to the starting point. We then use this minimum to generate a starting-point to find the minimum of $I_{\text{tot}}^{l+3}$ and then use that minimum to find the minimum of $I_{\text{tot}}^{l+4}$ etc. We stop when the $\lambda$-local density matrices gotten from minimizing $I_{\text{tot}}^{\lambda+1}$ is the same (up to the precision used) as the local density matrices gotten from minimizing $I_{\text{tot}}^\lambda$.

Since $I_{\text{tot}}^\lambda$ is not convex we cannot be sure that we have found the global minimum. However, in a region close to a maximally mixed set of density matrices the Hessian $\mathbf{H}_{I_{\text{tot}}^\lambda}$ is positive definite. So, one would expect that this would typically not be a problem. Furthermore, we know that $I_{\text{tot}}^\lambda$ is bounded from below by $\min(I_{\text{tot}}^{l+1})$ (the minimal value of $I_{\text{tot}}^{l+1}$) and that we can find with certainty. Then using Kim's inequality (148), this gives us a region in which the global minimum must be. For the local Gibbs state in Fig. 5 the difference between $\min(I_{\text{tot}}^{l+5})$ and $\min(I_{\text{tot}}^{l+1})$ is small,

$$\min(I_{\text{tot}}^{l+5}) - \min(I_{\text{tot}}^{l+1}) \approx 2.30 \times 10^{-9} \,. \tag{153}$$

(For $\lambda = l + 5$ the algorithm had converged to machine precision.) So unless $\mathbf{H}_{I_{\text{tot}}^\lambda}$, for some unknown reason, has some strongly oscillatory behavior, we can be certain that $\mathbf{H}_{I_{\text{tot}}^\lambda}$ is positive definite within a region which must contain the global minimum of $I_{\text{tot}}^\lambda$, and we can then be certain that we have found the global minimum.

## E.2 Finding the logarithm of an $l$-local Gibbs state

When we have found the $k$-local density matrices $\Omega_{\text{Gibbs}}^l$ ($k > l$) in an $l$-local Gibbs state $\rho_{\text{Gibbs}}^l$, we can use the result to also find the terms $\omega^l = \{\omega_n^l\}$ all $n$ of the operator $\mathcal{O} = \sum_n \omega_n^l$, which is the negative logarithm of the Gibbs state,

$$\rho_{\text{Gibbs}}^l = e^{-\mathcal{O}} \,. \tag{154}$$

There are in principle several ways to decompose the $\mathcal{O}$ into a set $\omega^l$. Any set with the property

$$\langle \omega^l | \mathbf{T}_{\to l} \varrho \rangle = \langle \mathcal{O} | \varrho \rangle \,, \tag{155}$$

for all Hermitian matrices $\varrho$ on the full space will do. So $\omega^l$ is only defined up to an arbitrary element in $\perp \text{im}(\mathbf{T}_{\to l})$. If we assume that the algorithm described in the previous subsection converged at stage $\lambda$, then this means that

$$I_{tot} - I_{tot}^\lambda = 0 \,, \tag{156}$$

up to the precision used. Since $I_{tot} - I_{tot}^\lambda$ is non-negative its gradient thus must vanish, from which it follows that

$$\mathcal{O} = -\nabla I_{tot} + \mathbb{1} = -\nabla I_{tot}^\lambda + \mathbb{1} = \sum_n' \log_2(\Omega_n^{\lambda-1}) - \sum_n \{\log_2(\Omega_n^\lambda) \,, \tag{157}$$

where as before the sum $\sum_n'$ indicates that the sum runs over all $n$ except the ones corresponding to the left and the right most neighborhoods. In the last equality we used the rewriting of the formula

$$\mathcal{I}_{\text{tot}}^l = \sum_n I(\Omega_n^l) - \sum_n' I(\Omega_n^{l-1}), \tag{158}$$

explained in Fig. 9 and the expression $\nabla I(\Omega_n^l) = \log_2(\Omega_n^l) + \mathbb{1}$. For an arbitrary Hermitian matrix $\varrho$ on the entire space we then get

$$\langle\mathcal{O}|\varrho\rangle = \left\langle \{(1-\delta_{n,n^{\text{right}}})\mathbb{1}_{n-l/2} \otimes \log_2(\Omega_{n+1/2}^{\lambda-1}) - \log_2(\Omega_n^\lambda)\}_{\text{all } n} \middle| \mathbf{T}_{\to\lambda}\varrho \right\rangle, \tag{159}$$

where $n^{\text{right}}$ labels the rightmost scale-$\lambda$ neighborhood. Since $\mathcal{O} \in \perp \ker \mathbf{T}_{\to l}$ this is equivalent to

$$\langle\mathcal{O}|\varrho\rangle = \left\langle \mathbf{T}_{\lambda\to l}^+ \mathbf{T}_{\lambda\to l} \times \{(1-\delta_{n,n^{\text{right}}})\mathbb{1}_{n-l/2} \otimes \log_2(\Omega_{n+1/2}^{\lambda-1}) - \log_2(\Omega_n^\lambda)\}_{\text{all } n} \middle| \mathbf{T}_{\to\lambda}\varrho \right\rangle. \tag{160}$$

Furthermore, it can be shown that when acting on elements in $\text{im}(T)$

$$\mathbf{T}_{l\to l'}^+ = \frac{N-l}{N-l'} d^{l-l'} \mathbf{T}_{l\to l'}^T, \tag{161}$$

where $N$ is the total number of sites. Using this expression in the previous equation we get

$$\langle\mathcal{O}|\varrho\rangle = \frac{N-l}{N-l'} d^{l-l'} \left\langle \mathbf{T}_{\lambda\to l} \middle| \{(1-\delta_{n,n^{\text{last}}})\mathbb{1}_{n-l/2} \otimes \log_2(\Omega_{n+1/2}^{\lambda-1}) - \log_2(\Omega_n^\lambda)\}_{\text{all } n} \middle| \mathbf{T}_{\to l}\varrho \right\rangle. \tag{162}$$

Comparing with (155) it then follows that

$$\omega^l = \frac{N-l}{N-l'} d^{l-l'} \mathbf{T}_{\lambda\to l} \left\{ (1-\delta_{n,n^{\text{last}}})\mathbb{1}_{n-l/2} \otimes \log_2(\Omega_{n+1/2}^{\lambda-1}) - \log_2(\Omega_n^\lambda) \right\}_{\text{all } n} \tag{163}$$

is a decomposition of $\mathcal{O}$. In fact, since it is an element of $\text{im}(\mathbf{T}_{\to l})$, it follows that it is the unique minimum norm decomposition.

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
