# Peer review of "Time-evolution of local information: thermalization dynamics of local observables"

_SciPost Physics, doi:SciPost Phys. 13, 080 (2022)_

## Round 1 · Referee Report · Anonymous (Referee 1) · 2021-8-11

Report

The purpose of the manuscript is two-fold: first, the authors introduce a concept for characterizing the evolution of quantum information in far-from-equilibrium settings (the so-called information lattice) and study its behavior in a paradigmatic example. In the second part, they build on the intuition gained from this to propose a numerical method for simulating such quantum dynamics with limited classical resources, which they benchmark by calculating a diffusion constant in the same model.

I found the paper quite interesting. The information lattice is a nice and intuitive way of picturing the flow of quantum information from small to large scales and the emergence of thermal states. The proposed numerical technique is certainly very timely, with several other works proposing methods to accomplish similar goals in recent years. For these reasons, I think the manuscript is very well suited for publication in SciPost. However, there are various questions and comments I have that I think should be addressed before publication.

1) I found some parts of the paper somewhat difficult to read. In particular, the introduction is quite long and hard to follow, since the relevant definitions are only introduced later on. Indeed, certain parts of it are later repeated almost verbatim.

Also, the technical derivation in App. A is hard to follow, with various notations introduced. While I understand that this might be unavoidable, I think it would be beneficial for the readers to have a shorter summary of what exactly are the steps of the algorithm.

2) The authors claim at several places that under the unitary dynamics, information only ever flows from smaller to larger scales, without any 'backflow'. If this was the case, then in principle it would be possible to simulate local properties exactly with very limited resources (keeping only subsystems up to l=2,3 say). It seems unlikely to me that this is true. Indeed, such backflow processes are mentioned e.g. in arXiv 1710.09835. This question, and its relevance for the proposed algorithm, should be discussed, at least briefly.

3) In the manuscript, only a single model is considered, taken from Ref. 46. My understanding is that this particular model might be quite special - in Ref. 46 it was found that it is possible to accurately simulate its dynamics using TDVP, while this is not true in general, as was shown later in arXiv 1710.09378. This raises the question whether the rapid convergence seen in the present manuscript might also be due to some particular properties of this model. On a related note: in Ref. 46, the best estimate of the diffusion constant seems to be around D=0.55. This is different from the result claimed here (D=0.45) by about 20%. Do the authors have a proposed resolution of this discrepancy?

4) While the authors admit that the condition in Eq. (60) is only heuristic, I would have found it useful to have a slightly more detailed discussion of why one would expect it to be a reasonably good approximation, or indeed to test it numerically (I also do not understand the analogy with Fick's law).

Relatedly, it seems to me that for longer range interactions, one would need to supplement it with additional conditions for the longer range currents (J_{l->l+2} etc), so it is not immediately clear how to apply the method to these cases.

Some smaller comments:

5) The discussion of 'local equilibrium' in the paper is somewhat confusing. In the manuscript, it is defined by the requirement of vanishing information current at some scale. However, I would expect that the current never exactly vanishes (indeed, the authors themselves say this in a footnote), so it is unclear what the precise definition should be, and whether the distinction between finite/infinite local equilibration time is meaningful. In particular, even for a translationally invariant state, one expects a slow, power-law approach towards the thermal state (see e.g. arXiv 1311.7644). Would the authors expect this not to show up in the quantities they consider?

6) I found the discussion of the relationship between the new numerical method proposed and the existing literature somewhat lacking. It is claimed that previous methods underestimate the information current at some intermediate scale. But it is unclear why this is an issue, if we take at face value the claim that there is no flow of information from these scales back to the smaller scales of interest (see point (2) above). It is also not clear if this claim is even true of all the previous methods mentioned. In particular, the one in Ref. 48 is very similar in spirit to the one proposed here.

7) Some typos: 'full-filling', 'full filled', 'Hilbertspace', 'below expressions'

  • validity: -
  • significance: -
  • originality: -
  • clarity: -
  • formatting: -
  • grammar: -

Author:  Thomas Klein Kvorning  on 2022-06-15  [id 2582]

(in reply to Report 1 on 2021-08-11)
Category:
answer to question

We are happy that the referee finds our article “quite interesting”. We are thankful to the referee for finding errors and pointing out the parts of the article which can be misunderstood. We have made a major revision of the presentation and it hopefully now should be more clear.

1a) I found some parts of the paper somewhat difficult to read. In particular, the introduction is quite long and hard to follow, since the relevant definitions are only introduced later on. Indeed, certain parts of it are later repeated almost verbatim.

We made a significant revision to resolve these issues.

1b) Also, the technical derivation in App. A is hard to follow, with various notations introduced. While I understand that this might be unavoidable, I think it would be beneficial for the readers to have a shorter summary of what exactly are the steps of the algorithm.

In App. A we introduce some technical notation needed to make some of the technical derivations in the preceding appendices less involved. We explain the algorithm in the main text of the paper. However, in App. B, we present how to effectively perform the minimization of Taylor expansion of the total information. One could consider it part of the algorithm since if one, e.g., naively constructed the entire Hessian matrix and inverted it, the truncation values that one could reach would be much smaller than we use. However, since this optimization is not needed to define the algorithm and is somewhat technical, we thought it more suitable for an Appendix.

2) The authors claim at several places that under the unitary dynamics, information only ever flows from smaller to larger scales, without any 'backflow'. If this was the case, then in principle it would be possible to simulate local properties exactly with very limited resources (keeping only subsystems up to l=2,3 say). It seems unlikely to me that this is true. Indeed, such backflow processes are mentioned e.g. in arXiv 1710.09835. This question, and its relevance for the proposed algorithm, should be discussed, at least briefly.

We apologize if we have been unclear. We assume that the general flow is from small to large scales and that the effect of correlation on large scales rapidly decays with scale. Information can return from one scale to a slightly smaller one. One can see such examples in our data. We made this more apparent in the current manuscript.

3a) In the manuscript, only a single model is considered, taken from Ref. 46. My understanding is that this particular model might be quite special - in Ref. 46 it was found that it is possible to accurately simulate its dynamics using TDVP, while this is not true in general, as was shown later in arXiv 1710.09378. This raises the question whether the rapid convergence seen in the present manuscript might also be due to some particular properties of this model.

We did not want to make the paper even longer by presenting data for several different models. Wether the Matrix-product-state time-dependent variational principle (MPSTDVP) works is not a yes/no question. As the referee mentions, the estimate given by MPSTDVP from ref. 10 is roughly in the range D=0.5 to 0.55. These are only the fluctuations in the late time approximation of the diffusion coefficient (the diffusion constant); the worst finite time estimates of the diffusion coefficient have much larger fluctuations. The chosen model is not fine-tuned to work particularly well with MPSTDVP, and there are situations where MPSTDVP works better and situations where it works worse. As the referee mentioned, there are also situations where MPSTDVP works much worse when the diffusion coefficient plateau to seemingly one diffusion constant, and then at a later time (which is bond-dimension dependent), the diffusion coefficient changes to another plateau with a value that can differ by more than 100%. The conclusion we draw is that it was somewhat “lucky” in ref. 10 that the MPTDVP algorithm works reasonably well. However, it is by no means a fine-tuned point.

Nevertheless, to ease the referees' worry, let us share some data for another model used in the literature. Here we consider the mixed field Ising Hamiltonian from Ref. 12 (arxiv:2004.05177, ref 48 in the old manuscript), studied with the dissipation-assisted operator evolution method (in the attached data, we use the same conventions and consider the same simulation as there). We attach a figure with truncation values l_c=6,7,8 (we skipped the largest truncation value we use in the article). Convergence is very similar to the model we consider in the article (but slightly faster since we reach less than 1% difference between the largest truncation variables at one truncation value smaller). For easy comparison, the plot is made with the same scale as Figure 3c in Ref. 12.

After carefully examining the data in Ref. 12, our data seem to agree. Ref. 12 defines the diffusion constant as the average in the time-range 15-20. Let's call this average D. (This underestimates the diffusion constant since it's still increasing at t=15, but we use this definition to compare.) They have two truncation variables, l and \gamma, and their algorithm is exact in the limit of either l\rightarrow\infty or \gamma\rightarrow0. To get a value for D, they linearly extrapolate D(\gamma) to D(0) for different values of l. See attached Figure arxiv2004.05177Fig3a, taken from ref 12 (but with the thin dashed lines added by us). Since they do a linear extrapolation to \gamma=0 and D(\gamma) has a positive second derivative, they have underestimated the diffusion constant (see the aforementioned figure). Instead, a quadratic extrapolation gives diffusion constants in the range [1.41,1.43] (the thin dashed lines added by us in the figure). This agrees with the value gotten from our largest truncation value, 1.42, which converged to 1%. (We converge from above, so our value is an overestimation. An exponential extrapolation where we assume the relative difference continues to decrease at the same rate as for the two largest pairs of values gives the value D=1.411. The data set from Ref. 12, which seems most reasonable to extrapolate, is for l=4. The quadratic extrapolation of that dataset gives D=1.409, 0.1% smaller than our estimate, well within tolerance.)

3b) On a related note: in Ref. 46, the best estimate of the diffusion constant seems to be around D=0.55. This is different from the result claimed here (D=0.45) by about 20%. Do the authors have a proposed resolution of this discrepancy?

We are not worried that our estimate of the diffusion constant is almost 20% smaller than the MPTDVP result. The fluctuations with the bond dimension of the MPTDVP estimate of the diffusion coefficient at late times (i.e., the diffusion constant) converge to less than 10%. However, the most considerable fluctuations at intermediate times are much more significant than 20%. So the MPTDVP estimate could be 20% wrong.

4) While the authors admit that the condition in Eq. (60) is only heuristic, I would have found it useful to have a slightly more detailed discussion of why one would expect it to be a reasonably good approximation, or indeed to test it numerically (I also do not understand the analogy with Fick's law).

Relatedly, it seems to me that for longer range interactions, one would need to supplement it with additional conditions for the longer range currents (J_{l->l+2} etc), so it is not immediately clear how to apply the method to these cases.

First, the analogy with Fick’s law was unfortunate and based on indirect reasoning. We agree that it is not clear and have removed it from the updated manuscript.

The heuristic boundary condition is based on the idea that the relative change in the information in layer l and layer l+1 should be roughly the same during a small time interval.

A similar condition with a more extended range Hamiltonian can also be used, but it is correct as the referee states that the precise expression would look different.

5a) The discussion of 'local equilibrium' in the paper is somewhat confusing. In the manuscript, it is defined by the requirement of vanishing information current at some scale.

Yes, we agree that the term local equilibrium, which we used, could lead to confusion, so we renamed it. Thank you for the comment.

5b) However, I would expect that the current never exactly vanishes (indeed, the authors themselves say this in a footnote), so it is unclear what the precise definition should be, and whether the distinction between finite/infinite local equilibration time is meaningful. In particular, even for a translationally invariant state, one expects a slow, power-law approach towards the thermal state (see e.g. arXiv 1311.7644). Would the authors expect this not to show up in the quantities they consider?

As the referee mentions, with a translation invariant Hamiltonian and a homogenous initial state still, the local density matrices, generically, have corrections that have a power-law approach to the equilibrium values. It was a mistake to suggest otherwise. We have added a clarifying paragraph.

Concerning the current exactly vanishing, it, of course, never does. The question is if it decays fast enough that an l-local Gibbs state can (up to some tolerance) be used to calculate the derivatives of the l-local density matrices.

6a) I found the discussion of the relationship between the new numerical method proposed and the existing literature somewhat lacking. It is claimed that previous methods underestimate the information current at some intermediate scale. But it is unclear why this is an issue, if we take at face value the claim that there is no flow of information from these scales back to the smaller scales of interest (see point (2) above).

Our line of reasoning is that if the current is consistently severely underestimated, the erroneous buildup would eventually become significantly larger than the information on small scales that we try to capture. So even if a tiny fraction returns, the estimates of the local observables could be very skewed. Nevertheless, suppose the influence of the correlations on a scale l_c on small scales decrease exponentially as l_c increase. In that case, this buildup of information will eventually (as the referee argues) for large enough l_c not be an issue. The point is that with any algorithm, one can only reach modest values of l_c, so this slowdown in convergence is a problem.

6b) It is also not clear if this claim is even true of all the previous methods mentioned. In particular, the one in Ref. 48 is very similar in spirit to the one proposed here. (Author comment: Ref 48 is Ref 12 in the new manuscript (arxiv:2004.05177))

In the discussion we think the referee referred to, we wanted to make the point that a time-evolution algorithm where one repeatedly approximates using states which have correlations decaying exponentially for scales larger than a value l* (given by the truncation value) will lead to underestimation of information currents at that scale. (This is, e.g., true for MPS and MPOs.) The reason is that such states will necessarily also have exponentially decaying information current. Therefore, the information current will be underestimated, leading to a buildup of erroneous information.

As stated by the referee, the dissipation-assisted operator evolution method from Ref. 12 can be expected to (at least partially) overcome this issue. To understand why let us give a simplified picture of that algorithm. It is based on repeatedly time-evolving the state quasi-exactly for some time \Delta t and then approximating by a smaller bond dimension MPO but preserving the l smallest density matrices. After that approximate step, the information current is most likely underestimated at l, but one would expect it relatively fast to return to the correct value. So if \Delta t is large, the time-averaged information current would be roughly correct. It was unfortunate that our previous passage implied otherwise.

Attachment:

Data.pdf

---

## Round 1 · Referee Report · Anonymous (Referee 2) · 2021-10-18

Report

The presented ideas on information flow in quantum many-body systems are very interesting. Based on these, a simulation technique is introduced to only evolve reduced density matrices for small subsystems while avoiding any reference to a global state. While the proposed ideas and conjectures about many-body dynamics are very interesting, at this point, there appear to be gaps in the justification and numerical substantiation. For now, I don't see that the paper reaches the level of "groundbreaking results" that SciPost Physics aims for.

The approach is centered around the von Neumann information of subsystem states with a hierarchy of larger and larger subsystems. Using the mutual information, in a kind of cluster expansion, the total von Neumann information is written as a sum of mutual informations on subsystems of increasing sizes. It is then studied how these mutual informations evolve in time. As the total information is conserved, one can also define information currents. Arguing that information should generally flow from small to large scales, it is concluded that the dynamics on short scales should decouple from the dynamics on larger scales and a corresponding simulation technique on subsystem density matrices is suggested. The technique is tested for a certain high-energy state of the Ising chain with transverse and longitudinal fields. Results for truncations of the hierarchy at different maximum length scales l are compared to l=9 and convergence of the l<9 results to the l=9 results is observed.

I have some questions on the concepts that don't appear to be answered in the current version of the manuscript:

1) It is unclear why a decay of information flow between different length scales would imply an uncoupling of the corresponding dynamics. If we observe that mutual informations for small subsystems equilibrate, why should this imply a decoupling of the local dynamics from that on larger length scales?

2) It is stated multiple times that information should generally flow from small length scales to larger length scales. It is likely that one can specify scenarios where this behavior occurs, but the paper does not attempt to give specific preconditions or to give a proof. For what it's worth, nothing keeps us from considering, for example, a highly entangled initial state psi(t0) which has been obtained by evolution under Hamiltonian H starting from a product state. Evolving this state with -H would lead to a decay of long-range entanglement for t=0 to t0. Revival scenarios that have been discussed theoretically and observed in experiments should correspond to cases where "information" flows forth and back between different length scales.

3) An important role is hence played by the truncation scheme. The authors suggest Petz recovery maps and employing Gibbs states that are compatible with the density matrices of small subsystems. It is not obvious why the steady states that are obtained in this way would not decisively depend on the truncation scheme and what properties a truncation scheme should have in order to obstruct the evolution on small scales as little as possible.

4) The paper does not address the N-representability problem: Given local states, it is unclear whether a compatible global state exists at all. In fact, the N-representability problem is known to be QMA complete. Hence, the proposed "Gibbs state defined by \Omega^l" (the l-site subsystem density matrices) may not exist. One may only be able to find a best approximation.

The benchmark simulations are done for an Ising chain and an initial state that is the identity except for a single site. Results for truncation at l<9 are compared to the result for l=9. The shown observables are the diffusion constant and a single-site magnetization. Seeing that deviations of l=7 are smaller than those of l=8 etc. is not conclusive. To make a convincing argument about convergence, it would be advisable to compare against an independent quasi-exact simulation technique. For short times, this could be done with exact diagonalization or time-dependent DMRG. Given that some assumptions about the dynamics, that are needed for the approach to work, are difficult to prove, it would be very useful to show data for further models like the Heisenberg chain and/or the Hubbard model. It would also be helpful to see the performance for initial states other than the employed high-energy product state.

Some minor comments on the presentation: - The introduction and other parts of the text appear a bit lengthy and repetitive. Conjectures about properties of the many-body dynamics like the dominance of information flow to larger scales are repeated a number of times. - There are some minor orthographic mistakes like "in a course grained picture", "note that care most be taken", "Hilbertspace". - Some notations are unusual like the "n" below "Tr" in Eq. (51) or "l>l'<L" on page 13. Also, I'd interpret "\rho_{[n,n+l]}" as an l+1 site density matrix instead of the intended l-site density matrices. - The word "information" is apparently used with different meanings: the mutual information and the subsystem density matrices. This can be confusing at times. - Some equation references seem mistaken. For example, the caption of Figure 7 refers to Eq. (53) as the initial state.

  • validity: ok
  • significance: high
  • originality: high
  • clarity: ok
  • formatting: good
  • grammar: good

Author:  Thomas Klein Kvorning  on 2022-06-15  [id 2581]

(in reply to Report 2 on 2021-10-18)
Category:
answer to question

First, we apologize for the slow response time and thank the referee for commenting and reading our manuscript.

We are glad to hear that the referee finds our ideas "very interesting" but are sorry that they still think the paper does not meet the criteria for publication in SciPost. By reading the referees' comments, we have concluded that the recommendation of not publishing is based partly on misunderstanding some of the material and partly on misunderstanding the purpose of this article. We have made an extensive rewriting of the article to make it more pedagogical and hope this will avoid confusion in the future.

This article aims to introduce the information lattice as a tool to diagnose many-body quantum dynamics. To show that this tool is valuable, we show that the intuition gained from using it leads to a development of an algorithm to capture the evolution of local observables without also simulating longer range correlations. There is vast literature on trying to accomplish such algorithms (see references in the article). The information lattice exposes weaknesses in previous attempts and, as we show, also points to a way forward in overcoming these weaknesses. Answer to the questions.

1) It is unclear why a decay of information flow between different length scales would imply an uncoupling of the corresponding dynamics. If we observe that mutual informations for small subsystems equilibrate, why should this imply a decoupling of the local dynamics from that on larger length scales?

If the system has reached equilibrium, the local density matrices have decoupled from long-range correlations. That is the basis of quantum statistical mechanics. We generalize this in the article. Suppose the information on some scale vanishes, say at L. In that case, one can calculate the derivative of the local density matrices on scale L-1 without reference to any correlations on longer scales. These local density matrices can thus be time-evolved without knowing correlations on larger scales.

2) It is stated multiple times that information should generally flow from small length scales to larger length scales. It is >likely that one can specify scenarios where this behavior occurs, but the paper does not attempt to give specific >preconditions or to give a proof. For what it's worth, nothing keeps us from considering, for example, a highly entangled >initial state psi(t0) which has been obtained by evolution under Hamiltonian H starting from a product state. Evolving >this state with -H would lead to a decay of long-range entanglement for t=0 to t0. Revival scenarios that have been >discussed theoretically and observed in experiments should correspond to cases where "information" flows forth and >back between different length scales.

A simple counting argument will tell you that the number of constraints you have to put on a state to get the same amount of information on scale l grows exponentially as l decreases. This counting argument is the basis for why information generically flows from small scales to large. Our algorithms rely on this assumption and will surely fail if it is not fulfilled. Therefore, we cannot capture the phenomena the referee mentioned in their question. But that has never been our intent; the point we (and several others referenced in the article) are making is that you can capture the time-evolution of local observables in typical time-evolutions with small resources. Still, of course, one cannot tackle the general case of quantum many-body time-evolution.

Let us discuss the two examples the referee mentioned in more detail to see why they are irrelevant.

First, quantum revivals in finite systems indeed cannot be captured by our algorithm. However, revival depends on a maximal scale to which information can flow. The premises for our work is large systems where information can escape to arbitrarily large scales. By construction, we cannot study finite-size corrections.

The referee's second example is a local state that is time-evolving with a generic local Hamiltonian for a long time T, after which the Hamiltonian is time-reversed. For this example, our algorithm would surely fail. However, this example is fine-tuned: only a minimal perturbation of the Hamiltonian after the time-inversion will generically mean that information will not return to the smallest scales. A classical analog of this example is an initial state where all molecules in a room are contained within a small volume. Using the second law of thermodynamics, one can conclude that the air will soon be roughly homogenous. However, if the Hamiltonian were then time-reversed, the entropy would rapidly decrease, and all molecules would again collect in a small volume. Does this mean that we should abandon the second law of thermodynamics?

3) An important role is hence played by the truncation scheme. The authors suggest Petz recovery maps and employing >Gibbs states that are compatible with the density matrices of small subsystems. It is not obvious why the steady states >that are obtained in this way would not decisively depend on the truncation scheme and what properties a truncation >scheme should have in order to obstruct the evolution on small scales as little as possible.

In the paper, we say that if an information gap occurs in the information lattice, then the time-derivative of the local density matrices is unchanged if we assume the state to be a local Gibbs state. This is a true statement. (What cannot be guaranteed is that even if a gap opens, information from large scales does not at a later time return, and then this statement would no longer be accurate.)

If no such gap occurs, we do not have a solution to time-evolve the local density matrices with a definite bound on the error. In this case, we make the point that one cannot time-evolve the local density matrices by assuming that the state is a local Gibbs state.

The referee also asks why one at all would trust a truncation scheme controlled by convergence with a truncation variable. This is again asked in 5), so we postpone the answer to that question.

4) The paper does not address the N-representability problem: Given local states, it is unclear whether a compatible >global state exists at all. In fact, the N-representability problem is known to be QMA complete. Hence, the proposed >"Gibbs state defined by \Omega^l" (the l-site subsystem density matrices) may not exist. One may only be able to find >a best approximation.

First, is the N-representability problem relevant for the examples we consider in the article? No, it's not. To understand why, consider a typical state on a Hilbert-space with dimension D and consider reduced density matrices up to some precision \epsilon on dimension d<<D. Then it is generically not a hard problem to reconstruct a density matrix consistent with these small density matrices. Our examples are similar to this situation. We can find a generalized Gibbs state consistent with the local density matrices, so at least one compatible density matrix exists.

Second, can we find an l-local Gibbs state given any local density matrices, and would that not contradict the N-representability problem? No, we cannot, and yes, it would. As we explain, our algorithm only converges if there exists an l-local Gibbs state with a correlation length short compared to the maximal size of density matrices we can handle.

5) The benchmark simulations are done for an Ising chain and an initial state that is the identity except for a single site. >Results for truncation at l<9 are compared to the result for l=9. The shown observables are the diffusion constant and a >single-site magnetization. Seeing that deviations of l=7 are smaller than those of l=8 etc. is not conclusive. To make a convincing argument about >convergence, it would be advisable to compare against an independent quasi-exact simulation technique. For short >times, this could be done with exact diagonalization or time-dependent DMRG. Given that some assumptions about the >dynamics, that are needed for the approach to work, are difficult to prove, it would be very useful to show data for >further models like the Heisenberg chain and/or the Hubbard model. It would also be helpful to see the performance for >initial states other than the employed high-energy product state.

As we explain in the article, the time-evolution is quasi-exact (has a controlled bound on the error) for short times. The times one could reach with time-dependent DMRG where there is a reasonable small bound on the error are roughly the same as our algorithm. So we cannot see the point with that exercise.

We present a novel approach to simulate quantum many-body dynamics and exemplify it on a non-trivial used previously in the literature, in ref. 10. As opposed to, e.g., ref. 10 also employing uncontrolled truncation schemes, we show that we have convergence for all times, not just, e.g., the diffusion coefficient at infinite time, we also non-local observables such as the information currents. If you only have convergence at late times and a region of intermediate times where the algorithm where you have large fluctuations, then we can see why it can be hard to trust. However, with that said. To be fully confident that there, e.g., is not a slow drift of the values, we would have liked to see a convergence that continued a few orders of magnitude after a rough convergence of, say, 1% occurred. We agree that there is still room for objections to how well the data has converged, but we would like to point out that we know no algorithm where a higher degree of convergence has been seen with the same amount of numerical resources. (We are currently working on a new algorithm building on the ideas of this article, for which we expect to see much faster convergence.)

Concerning studying other time evolutions, we refer to the answer given to the other referees' questions.

---

## Round 2 · Referee Report · Anonymous (Referee 3) · 2022-6-18

Report

The reply, does not address my main concerns.

For example, my first point (1) was that it is unclear why a decay of information flow between different length scales would imply an uncoupling of the corresponding dynamics. If we observe that mutual information for small subsystems equilibrates, why should this imply a decoupling of the local dynamics from that on larger length scales?

The authors' answer reiterates the argument that, if information (flow) vanishes at a scale L, one could obtain the time derivative of the density matrices at scale L-1 without reference to larger scales. That is exactly the point in question. The information currents are just some scalars. Their vanishing does in itself not guarantee an uncoupling of dynamics (density operators). Yes, it would imply that the density-matrix time derivative as determined from the maximum entropy principle would not depend on larger-scale correlations, but under what constraints is the maximum entropy principle applicable in this way? The information currents alone do not provide a justification. While the approach may work under certain constraints, such an essential aspect of the proposal requires a more detailed reasoning and discussion of the required conditions.

The authors' state that the two simple counter examples given in my point (2) would not be covered by the method. One can easily come up further scenarios. What is missing is a criterion that tells us when the results of the proposed method are trustworthy. When truncating the hierarchy at distance L, what kind of control do we have on the error introduced due to that truncation? How does one decide whether, say, a certain quantum quench falls into the considered class of "typical time-evolutions". One needs a corresponding framework to make the approach predictive.

BTW, "quantum revivals" are not limited to finite systems. And no, I do not suggest to abandon the second law of thermodynamics. On the other hand, it certainly does not mean that entropy would strictly increase under all circumstances or provide a derivation for the suggested method.

My point (4) on the N-representability problem is discarded in the reply. Sure, if the local density operators correspond to a global Gibbs state (with a small deviation) than there is no question. But generally, we don't know when/if that condition is met. So generally, the method will lead to non-representable local density matrices (nonphysical states) - especially, if we have no criterion on the effect of the truncation.

In comment (5), I pointed out that observing L=6,7,8 results get gradually closer to the L=9 results does not imply that the dynamics is converged or even quasi-exact. Anything else would surely be troubling, but nothing assures us that the L=9 results are precise. This is quite different from MPS simulations, where the truncation error gives rigorous bounds on approximation errors, and the dynamics becomes exact for (very) large bond dimensions. To assess the accuracy, it seems imperative to compare against alternative quasi-exact methods.

There is still a fair number of grammatical mistakes, missing commas etc.

  • validity: -
  • significance: -
  • originality: -
  • clarity: -
  • formatting: -
  • grammar: -

Author:  Thomas Klein Kvorning  on 2022-07-05  [id 2632]

(in reply to Report 1 on 2022-06-18)
Category:
answer to question
reply to objection

Dear Reviewers,

First, we thank both reviewers for their quick reply to our long-delayed answer.

Many of the reviewer's comments seem to arise from a misconception of the scope of our algorithm. In particular, it seems that the reviewer has in mind generic matrix product state algorithms in $1D$ and is comparing only with those. These are essentially exact algorithms; as long as the bond dimension is large enough, one can make an exponentially small mistake on the full many-body wave function. This is useful in cases where there is not much entanglement in the state but breaks down when there is a lot of entanglement. This is the case in thermalizing dynamics, which is our focus. In this case, matrix product state algorithms can only exactly capture the full many-body dynamics to very short times. It should be clear that no classical algorithm will generally beat matrix product states; there might be specialized dynamics where one can find a more efficient algorithm but finding an algorithm that can exactly (meaning with small controllable error) capture all many-body quantum dynamics to long times is not likely going to happen. This would essentially mean that one could capture quantum many-body dynamics efficiently classically, and this we do not believe can be done.

There are special types of dynamics where people generally believe one can do better. Thermalizing dynamics is one such example. The reason for this is that, at long times, there is not much relevant information in the state, and one can effectively capture thermal states, for example, using purification via auxiliary degrees of freedom. But, in exact dynamics, one needs to keep track of all entanglement at all times, and at intermediate times this becomes impossible, and it's not possible to get to the long-time thermal state. That is, at least not without discarding some information. There is a lot of literature, which we cite, that aims to solve this problem. The second report of the other reviewer also explicitly acknowledges this outstanding problem.

We now have a less general but well-defined problem that we are trying to solve. One needs to discard some irrelevant information to solve this problem. We have introduced the information lattice as a tool to analyze which information to discard (incidentally, the information lattice is also a new contribution to the literature, and it is independent of the algorithm and can be used to analyze any quantum many-body dynamics). We have also discussed how we go about keeping track of relevant information and discarding irrelevant information. This is expected to work when the dynamics are thermalizing, in which case the irrelevant information that goes to large scales does not come back. Of course, there are many cases, some of which the referee has mentioned, where this does not hold. It does not hold in general that the information that goes to large scales will come back to short scales. But that is ok; those cases are by design not captured by our algorithm. One can compare this with the Boltzmann equation that correctly describes classical thermalization dynamics via the assumption of molecular chaos, which is essentially assuming one can discard higher-order correlations in scattering. This is, in some sense, an uncontrolled approximation, but the Boltzmann equation is still extremely useful. One can say the same about mean-field theory, which generally has uncontrolled approximations but is still very useful. In contrast, just as matrix product state algorithms, our algorithm has a controlled bound on the error in case of small entanglement, and when this bound is too large to be of interest, we still have some control, as we can vary the scale on which we keep information and compare. With this in mind, we answered the reviewer's questions in the first round, not spending too much time on those questions we felt were asking about cases beyond the scope of our work. Based on the second report of the reviewer, they have interpreted this as us trying to ignore the questions. We regret that our responses came across as such and attempt again to answer in more detail, but ask that the answers be read, keeping in mind that we are not suggesting that our algorithm can generally capture all many-body quantum dynamics to arbitrarily long times.

Our work is on, to quote reviewer 2, “the long-standing problem of simulating the dynamics of quantum thermalization,” and we hope that the reviewer agrees that on this problem we have made significant progress that deserves publication in SciPost Physics.

In the following, we will try to address in more detail the comments.

For example, my first point (1) was that it is unclear why a decay of information flow between different length scales would imply an uncoupling of the corresponding dynamics. If we observe that mutual information for small subsystems equilibrates, why should this imply a decoupling of the local dynamics from that on larger length scales? The authors' answer reiterates the argument that, if information (flow) vanishes at a scale L, one could obtain the time derivative of the density matrices at scale L-1 without reference to larger scales. That is exactly the point in question. The information currents are just some scalars. Their vanishing does in itself not guarantee an uncoupling of dynamics (density operators). Yes, it would imply that the density-matrix time derivative as determined from the maximum entropy principle would not depend on larger-scale correlations, but under what constraints is the maximum entropy principle applicable in this way? The information currents alone do not provide a justification. While the approach may work under certain constraints, such an essential aspect of the proposal requires a more detailed reasoning and discussion of the required conditions.

This point concerns the simplest case where we observe a vanishing of both values and flow of the information at some scale, i.e., the example detailed in Eq. 35 in Sec. 3. In that limit, we can use, e.g., the twisted Petz recovery maps given in Eq.10 to build the exact joint density matrix.

We emphasize that this is an exact result. With the assumption that information vanishes at some scale l, the density at that scale can exactly be reconstructed by the density matrices on scale l-1, and the matrices scale l-1 can therefore be time-evolved without reference to any larger density matrices. If the information is not exactly zero, we still have a controlled error since Eq.11 provides a bound on the error of the recovery map valid for finite information on scale l.

The requirement that information has vanished at a scale l at a certain point in time, t, does not mean that it necessarily will do so for all later time. It is easy to capture cases where information reaches l from smaller scales, but information on larger scales is lost, so we cannot guarantee that no information comes back from larger scales. So the technical requirement that information vanishes at some scale for all time t>t* is hard to verify. However, as discussed in the paper, based on the statistical drift of information towards larger scales, when information has vanished at a scale, information will generally not return to that scale from larger scales to where it has disappeared.

We emphasize that we do not consider examples where information does come back as uninteresting—quite the opposite—they are in the purest meaning of the word genuine quantum phenomena. An example is a collection of non-Abelian anyons moving by an external time-dependent potential. As long as they are well-separated, there is a scale l* larger than the coherence length $\xi$ where there is no information, and the local density matrices could be time-evolved exactly. However, when two anyone fuses an algorithm assuming no information on large scales would result in a local density matrix which is a mixture of all possible fusion products, even though the actual outcome is a definite fusion channel. This type of phenomenon is interesting but, unfortunately, out of the scope of this article.

The authors' state that the two simple counter examples given in my point (2) would not be covered by the method. One can easily come up further scenarios. What is missing is a criterion that tells us when the results of the proposed method are trustworthy. When truncating the hierarchy at distance L, what kind of control do we have on the error introduced due to that truncation? How does one decide whether, say, a certain quantum quench falls into the considered class of "typical time-evolutions". One needs a corresponding framework to make the approach predictive. BTW, "quantum revivals" are not limited to finite systems. And no, I do not suggest to abandon the second law of thermodynamics. On the other hand, it certainly does not mean that entropy would strictly increase under all circumstances or provide a derivation for the suggested method.

Indeed, quantum revivals can also happen in infinite systems in the presence, for example, of quantum many-body scars. We acknowledge that this is an example among many that are physically interesting and where information on large scales will come back and affect local observables, and our algorithm thus cannot work.

You, however, asked a different question. Are there situations where we can be sure our algorithms work? In the case of a scale l where information vanishes, we can prove our algorithm is exact (see the previous question). When information never exactly vanishes at a scale, we cannot prove that our algorithm works. Our guiding principle is that information only travels in one direction, from small to large scales, and it is in these situations where one can try our method. As you correctly point out, the controlled bound on the error in our algorithm is not helpful when there is no scale l for which the information is small. But we have a truncation variable $l_c$ for which we recover the exact result in the limit $l_c\rightarrow\infty$, and if we converge already for a finite $l_c$, it is indicative of having captured the exact time-evolution.

As we mentioned in the prolog to these answers, this kind of reasoning where an algorithm is motivated by a physical intuition but without rigorous boundaries to when one can apply it has been very successful in many areas of physics. Proving such strict bounds for our methods would clearly improve our work, but it is unreasonable to deem the work useless without it.

My point (4) on the N-representability problem is discarded in the reply. Sure, if the local density operators correspond to a global Gibbs state (with a small deviation) than there is no question. But generally, we don't know when/if that condition is met. So generally, the method will lead to non-representable local density matrices (nonphysical states) - especially, if we have no criterion on the effect of the truncation.

We did not mean to disregard your question; our answer was that the $N$-representability problem is not relevant for the time evolutions considered in the paper. We, however, acknowledge that we were lazy explaining this point last time and were far from pedagogical, for which we apologize.

In the numerical examples we consider, there exist $l$-local Gibbs states with short coherence length, which have the $l$-local density matrices as reduced density matrices. (This does not mean that the global state necessarily is a Gibbs state, just that there exists a Gibbs state, which is compatible). We verify this statement with our algorithm in App. E; in App. E.2 we show how one can find a set of operators on $l-1$ consecutive sites ${\omega_{n}^{l}}$ such that the density-matrix $\rho=\exp(\sum_{n}\omega_{n}^{l})$ has a given $l$-local density-matrix set as reduced density matrices. The algorithm only converges if there exists such a $\rho$ with a short coherence length for the given $l$-local density-matrix set. If such a global state does not exist, we generically can say nothing about the existence of a global density matrix. However, this is not the case for the numerical examples we consider.

The reason we do not discuss the $N$-representability problem in our paper is because we think such a discussion would draw attention away from the main points we are trying to convey. After all, it is not necessary to distinguish errors resulting in the $l$-local density matrices being incompatible with a global state and other errors. What matters is to estimate the error on the local density matrices. One can imagine density matrices incompatible with a global state but still only $\epsilon$ away from the correct local density matrices. In such a situation, these density matrices only give an $\epsilon$ error to any local observable. So what matters for the local observables is not whether a global compatible state exists but the error in the local density matrices.

As we discussed, in certain situations we do have a controlled bound on the error on the local density matrices and when we do not we control the error with the convergence as a function of our truncation variable $l_c$ (this of course has limitations as we previously discussed and you have pointed out).

In comment (5), I pointed out that observing L=6,7,8 results get gradually closer to the L=9 results does not imply that the dynamics is converged or even quasi-exact. Anything else would surely be troubling, but nothing assures us that the L=9 results are precise. This is quite different from MPS simulations, where the truncation error gives rigorous bounds on approximation errors, and the dynamics becomes exact for (very) large bond dimensions. To assess the accuracy, it seems imperative to compare against alternative quasi-exact methods.

Since we start out with states which only have information on scale $l=0$, it takes some time for the information on our truncation scale to become large. Until then we therefore have a small controlled bound on the error. The first simulation we do is quasi-exact for the time-range we show, so then there is no reason to compare with another method which also have a small controlled error (maybe except to show that our code is bug-free). For the second simulation, we only have a controlled error for early times in the simulation. However, using for example time-dependent DMRG to time-evolve we would also have a controlled error only for early times. So we cannot see the reason for such a comparison.

To summarize: in the time range where we can make a comparison with quasi-exact methods, our algorithm is also quasi-exact and the comparison would not be revealing. It would be valuable to go beyond this time range, but then there is nothing one can compare with.

There is still a fair number of grammatical mistakes, missing commas etc.

We have carefully proof-read the manuscript once again and hope we have caught most of the typos and grammatical errors.

---

## Round 2 · Referee Report · Anonymous (Referee 4) · 2022-6-20

Report

The authors have revised their manuscript, significantly clarifying its presentation. In their response, they have also provided additional data, which addresses concerns of fine-tuning I had of the original version. Given this, I believe their work provides a significant step forward in the long-standing problem of simulating the dynamics of quantum thermalization and is therefore appropriate for publication in SciPost Physics.

---

## Round 2 · Author Response

Dear Editor,

We thank you for organizing the review of our manuscript. We apologize for the delay in resubmitting our updated manuscript and the answers to the referees.

Since both referees complained about the accessibility of our manuscript, we have undertaken a significant rewriting of the article, particularly the first parts. We have significantly shortened the introduction, and instead of introducing background material in a separate section, we now rather introduce concepts when they are needed. We believe this has improved the paper's readability significantly and hope that the referees agree with this.

In addition, the first referee objected to publication based on what we think is a misunderstanding of the goal and contents of our paper. Our paper achieves two things: i) it introduces a way of separating quantum information into different scales—in what we call the information lattice—that gives a much more refined picture of the time evolution of quantum information than does, say, the entanglement entropy. Using this, we have analyzed generic quantum dynamics governed by a thermalizing Hamiltonian. And ii) using the insights from i), we suggest a new numerical algorithm that captures thermalizing dynamics by astutely throwing away quantum information that does not affect local observables. We show that this algorithm is competitive with the best available algorithms attempting to solve the same problem.

Given the above, we believe that our work makes significant progress on an open problem in the field of quantum many-body dynamics: how to simulate thermalizing dynamics for long times, given that thermal states have much less information than typical pure quantum states. There is an extensive literature trying to solve this problem in the last years (see references in our paper), and the referee seems to have missed this point. Instead, the objections to our algorithm are based on fine-tuned examples that obviously can not be captured by our algorithm, nor any other algorithms that attempt to simulate thermalizing dynamics to late times. It should be clear that no classical algorithm can capture all of the quantum information that is in a pure state time evolved for a long time unless it is for tiny systems that can be dealt with using exact diagonalization. In any case, we believe the above arguments should clarify that our paper satisfies both acceptance criteria 2 and 3 of SciPost Physics.

The second referee's main objection was on our choice of model since it may work better than expected for the matrix product state time-dependent variational principle. We have taken this seriously and have produced new data for a different model, directly comparing our results with the recent work arXiv:2004.05177, obtaining agreeing results (see details in answer to Ref. 2). We think this closes the worry that our data is somehow fine-tuned. Since it will not significantly add to the paper and anyway will be publicly available, we have decided to include the new data only in the response to the referee to avoid taking up much extra space in an already long article.

With these changes and our responses to the referees, we hope our manuscript can now be accepted for publication in SciPost Physics.

Yours sincerely,
Thomas Klein Kvorning
Loïc Herviou
Jens H Bardarson

---

## Round 2 · List of Changes

— Rewrote introduction
— Rewrote section 2
— Smaller changes and typo corrections throughout the manuscript.
— We also made several changes to our notation and nomenclature. The most major of these is that we removed the phrase local equilibrium since, as the second referee points out, our use of it can be confusing.

---

## Round 3 · Referee Report · Anonymous (Referee 2) · 2022-8-1

Report

I think that the authors' reply answered at least some of my concerns, and it would be valuable to include some of this discussion in the manuscript as it clarifies the scope and limitations of the method. With this, I recommend publication.

---

## Round 4 · Author Response

Dear Editor,

Thank you for your patience and time spent on our manuscript. We have now updated the manuscript to include the discussion with the second referee.

The main content in discussion with the referee not in the earlier manuscripts is the discussion on the N-representability problem. We previously omitted such a discussion since we thought it would distract attention from the article's main points. However, after the conversation with the referee, it was clear that it could be beneficial to discuss the subject. We added a paragraph in section 5, which includes the content from the discussion with the referee.

A related issue in the discussion is controlling the error in our algorithm. To make that more transparent, we rewrote parts of sections 4 and 5 to emphasize that we have a small controlled bound on the error under certain circumstances, and otherwise, we only control the error by convergence with the truncation variable.

One part of this discussion discussed the scope and aim of the article, which the referee initially misunderstood. This was already made significantly more pedagogical with the rewriting of our introduction after the first referee round, and the misunderstandings brought into the second round would not have occurred if the referee had read the new version first. Therefore we have not added any further discussion of this in the introduction.

With these changes, we hope the article will be accepted for publication in SciPost Physics.

---

## Editorial Decision

published